# The anatomy of past abrupt warmings recorded in Greenland ice

E. Capron [1,2✉], S. O. Rasmussen [1], T. J. Popp[1], T. Erhardt [3], H. Fischer [3], A. Landais[4], J. B. Pedro [1,5,6], G. Vettoretti [1], A. Grinsted[1], V. Gkinis [1], B. Vaughn[7], A. Svensson [1], B. M. Vinther[1] & J. W. C. White[7]

Data availability and temporal resolution make it challenging to unravel the anatomy (duration and temporal phasing) of the Last Glacial abrupt climate changes. Here, we address these limitations by investigating the anatomy of abrupt changes using sub-decadal-scale records from Greenland ice cores. We highlight the absence of a systematic pattern in the anatomy of abrupt changes as recorded in different ice parameters. This diversity in the sequence of changes seen in ice-core data is also observed in climate parameters derived from numerical simulations which exhibit self-sustained abrupt variability arising from internal atmosphere-ice-ocean interactions. Our analysis of two ice cores shows that the diversity of abrupt warming transitions represents variability inherent to the climate system and not archive-specific noise. Our results hint that during these abrupt events, it may not be possible to infer statistically-robust leads and lags between the different components of the climate system because of their tight coupling.

[1] Physics of Ice, Climate and Earth, Niels Bohr Institute, University of Copenhagen, Tagensvej 16, Copenhagen, Denmark. [2] Université Grenoble Alpes, CNRS, IRD, IGE, Grenoble, France. [3] Climate and Environmental Physics, Physics Institute & Oeschger Center for Climate Change Research, University of Bern, Sidlerstrasse 5, Bern, Switzerland. [4] Laboratoire des Sciences du Climat et de l'Environnement, LSCE/IPSL, CEA-CNRS-UVSQ, Université Paris-Saclay, Gif-sur-Yvette, France. [5] Australian Antarctic Division, Channel Highway, Kingston, TAS, Australia. [6] Australian Antarctic Program Partnership, Institute for Marine and Antarctic Studies, University of Tasmania, Hobart, TAS, Australia. [7] Institute of Arctic and Alpine Research, University of Colorado, Boulder, CO, USA. ✉email: emilie.capron@univ-grenoble-alpes.fr

Paleoclimatic records of the Last Glacial reveal a series of abrupt warming events occurring in the North Atlantic region, known as Dansgaard-Oeschger (D-O) events, with counterparts in lower latitudes[1] and Antarctic climate archives[2,3]. Oxygen isotope ($\delta^{18}$O) profiles from Greenland ice cores provide master records of this climate variability[4,5], illustrating fluctuations between Greenland Stadial (GS) phases with full glacial conditions and milder Greenland Interstadial (GI) phases (Fig. 1). The D-O climate variability is commonly linked to changes in the intensity of the Atlantic meridional overturning circulation (AMOC), resulting in heat transport changes from the low to the northern high latitudes[6,7]. However, no consensus exists yet to explain what triggers the abrupt warmings, characterized by Greenland surface temperature increases of 5–16 °C within a few decades to centuries[8]. Among the proposed paradigms, mechanisms involving changes in Nordic Seas sea-ice cover[9], atmospheric circulation[10], or the collapse of ice shelves[11] have been investigated. Recent studies suggest that abrupt climate variability can result entirely from unforced[12] or noise-induced oscillations of the coupled atmosphere-ice-ocean system that alter poleward energy transport (ref. [13] and [14] for reviews).

The mechanisms proposed to explain D-O event dynamics can be confronted with annual-to-decadal-scale observations of climatic changes across the globe over the GS–GI transitions. Indeed, such data sets provide a basis to map out the sequence of events, infer possible causal relations and evaluate hypothetical sets of governing mechanisms by comparing model output with the spatial expression and relative phasing of the observed changes, hereafter referred to as the "anatomy" of the changes. However, looking at the anatomy of abrupt events in paleoclimate data is challenging because it requires a high temporal resolution not attainable in most climatic archives, and because of relative dating uncertainties between paleoclimate records from different archives. Records of annual or close-to-annual resolution from Greenland ice cores overcome this challenge since they contain tracers recording conditions in different parts of the Earth System with each year's precipitation, all in one archive. The $\delta^{18}$O value of Greenland ice is mainly affected by local surface temperature changes, past changes in precipitation seasonality, the temperature at the moisture source regions, and elevation changes[15–18]. Hence, although $\delta^{18}$O is not a direct temperature proxy, it can be used as a qualitative tracer of local Greenland surface temperature changes. The second-order parameter d-excess (d-excess = $\delta$D − 8·$\delta^{18}$O) is commonly interpreted as a record of past changes in evaporation conditions or shifts in mid-latitude moisture sources[17,19,20], whereas $Ca^{2+}$ concentrations ($[Ca^{2+}]$) in Greenland ice cores reflect both source strength and transport conditions from terrestrial sources, which are mainly the mid-latitude Asian deserts[21,22]. Finally, changes in $Na^+$ concentrations ($[Na^+]$) can be interpreted as qualitative indicators of the sea-ice cover extent in the North Atlantic at the stadial-interstadial scale[23], whereas relative site accumulation rate changes can be estimated from the annual-layer thickness (denoted $\lambda$)[24]. Hence, ice-core multi-tracer studies are well suited to evaluate the precise phasing and duration of changes between different regions without relative dating uncertainty as all records come from the same core.

This approach was initially applied to characterize the sequence of events at the onsets of the Holocene, GI-1e (Bølling), and GI-8c[25,26]. For each of those transitions, a lead of a few years in changes in terrestrial aerosol concentrations, accumulation rate, and mid-latitude moisture sources relative to the changes in marine aerosols and the isotopic temperature was found. Such results suggest that the Greenland surface warming was preceded by changes in the conditions at the dust sources or changes to the transport to Greenland (e.g., rainfall-driven changes in aerosol washout). In parallel, the phasing between the high- and lower-latitude climate responses was investigated using ice-core

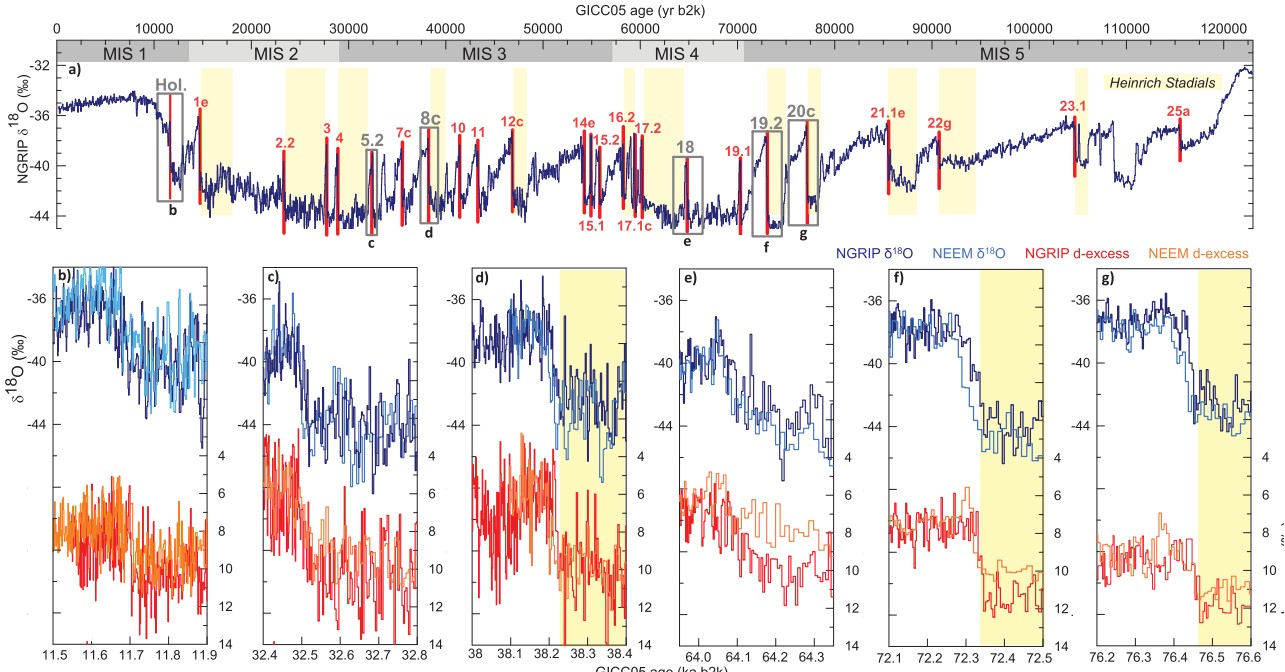

**Fig. 1 Abrupt climate variability recorded in Greenland water isotopic records. a** NGRIP $\delta^{18}$O record[5]. Studied abrupt warming transitions are highlighted with red vertical bars and Greenland Interstadials (GI) are numbered[38]. Gray boxes indicate intervals shown in (**b**–**g**), illustrating the variety of abrupt GS–GI transitions across the Last Glacial; stadials containing Heinrich events are indicated in yellow following refs. [53,85], and Marine Isotope Stages (MIS) are indicated in gray. **b**–**g** High-resolution $\delta^{18}$O from NGRIP (dark blue) and NEEM (light blue) and d-excess from NGRIP (red) and NEEM (orange) over 400 yr time intervals centered on the Holocene abrupt onset (**b**) and the abrupt transitions into GI-5.2 (**c**), GI-8c (**d**), GI-18 (**e**), GI-19.2 (**f**), and GI-20c (**g**).

gas-phase measurements: the $\delta^{15}N$ of $N_2$ as a tracer for Greenland surface temperature changes[27,28] and the methane concentration ($CH_4$) as a proxy for tropical climate change[29,30]. Although the first studies[29,31] estimated a lag of a few decades of tropical $CH_4$ emissions behind $\delta^{15}N$ at the onset of the abrupt warmings, a more recent study[32], focusing on the Bølling transition and using 5-yr-resolution $\delta^{15}N$ and $CH_4$ records, estimated that high- and low-latitude climate changes occurred essentially synchronously at that time, with Greenland surface temperature leading atmospheric $CH_4$ emissions by $4.5^{+21}_{-24}$ yrs, in agreement within errors with ref. [25].

Benefiting from the new NGRIP and NEEM high-resolution ice-core data sets, recent work extended the multi-tracer approach developed by ref. [25] and [26] to all transitions back to 60 ka b2k (thousand years before 2000 C.E.) and derived an average sequence of changes characteristic of the GI onsets by combining the estimated leads and lags for all studied transitions[23]. Based on the assumption that the relative timing differences between different tracers at all GI onsets are the result of the same underlying process, it was found that changes in both local precipitation and terrestrial dust aerosol concentrations led the change in sea-salt aerosol concentrations and $\delta^{18}O$ of the ice by about a decade. Event-stacking-based approaches are often applied to extract the common signal from highly variable climatic records[33–36]. Although this is useful, it is also worth looking into the details of the sequence of changes over each event, especially considering the high diversity observed in the amplitude of the warming[8], the shape and duration of GS and GI[37,38] (Fig. 1), and the evolving climatic background state throughout the Glacial (orbital configuration, global ice volume, and atmospheric greenhouse gas concentrations). Taking this view, we observe that the results from ref. [23] illustrate a decadal-scale range in leads and lags from one event to the next when considering the onset of each individual transition. These differences can be interpreted as coming from different realizations of the same set of underlying mechanisms owing to noise processes in the archive and internal variability in the climate system, or alternatively as a suggestion that one common set of mechanisms or sequence of events may not adequately describe the processes of all rapid warming transitions.

The aim of this study is twofold. First, we investigate the anatomy of the D-O warming transitions down to 112 ka b2k using a multi-tracer approach relying on new and existing records from the Greenland NEEM (77.45°N, 51.08°W) and NGRIP (75°N, 42.3°W) ice cores. Having so many highly resolved ice-core records from two different locations over numerous D-O events provides the most comprehensive opportunity so far to assess the geographical representativeness of single ice-core records. Second, the anatomy of D-O warmings inferred from Greenland ice-core data is compared with new simulations from the coupled Community Climate System Model Version 4 (CCSM4) as the basis for discussing the processes involved in D-O warmings.

We use here new and existing water isotope measurements ($\delta^{18}O$, d-excess) at high resolution (5 cm) from the NGRIP ice core[5] (Supplementary Data 1). The temporal resolution of the measurements corresponds to 1, 3, 4, 5 yr per sample at 10, 45, 80, and 105 ka b2k, respectively. We also include in our analysis, sections from the recent NEEM high-resolution water isotope records[39] for which the 5 cm resolution corresponds to 1, 4, 7, 18 yr per sample at 10, 45, 80, and 105 ka b2k. We also present high-resolution NGRIP and NEEM [$Ca^{2+}$] and [$Na^+$] records annually interpolated and extended back to ~108 ka b2k (Methods, Supplementary Data 2). Finally, we use the NGRIP $\lambda$ record back to 60 ka b2k obtained from the GICC05 annual-layer counting based on aerosol and visual stratigraphy records (Supplementary Data 3). We restrict our $\lambda$ analyses to the last 60 ka as

$\lambda$ is modeled from the stable water isotope record below this age and, therefore, is not independent of $\delta^{18}O$. The GICC05 chronology is applied to NEEM by means of interpolation between reference horizons of mainly volcanic origin[40]. The NEEM annual-layer thicknesses are only available as averages between these unevenly spaced reference horizons, rendering the NEEM $\lambda$ record unsuitable for this study. The NGRIP and NEEM data sets are reported on the GICC05 chronology back to 60 ka b2k and on the flow model-extended GICC05modelext chronology below this[40,41]. Age interpolation uncertainties limit the direct comparison of the absolute timing of changes between cores[40].

We use a probabilistic characterization of the transitions to infer the timing, duration, and amplitude of the local and regional changes associated with each studied D-O warming. Following refs. [23,25], we determine the relative phasing of changes in the different data sets by fitting a ramp (i.e., a linear change in the raw or logarithmically-transformed data between two stable states) to each data series within a prescribed search interval across each GS–GI transition (Supplementary Figure 1, Supplementary Table 1, Supplementary Data 4). We describe the ramp by the temporal midpoint of the ramp, the duration of the transition, the data value before the transition, and the amplitude of the change. Our probabilistic model also accounts for additive noise with autocorrelation (Methods). Note that our method is conceptually similar to ref. [23] with only minor differences in the parameter priors, whereas the uncertainty estimation is different from that employed by ref. [25], which used the RAMPFIT method[42]. In the following, we only display results for transitions where the ramp-fitting technique provides an unequivocal solution, i.e., the timing and duration of the identified onset and end of the transitions do not change by more than a decade when the width of the search time window is varied (Methods, Supplementary Figure 3).

## Results

To start, we describe the general characteristics of NGRIP and NEEM water isotopic records. Fig. 1 displays NGRIP and NEEM high-resolution $\delta^{18}O$ and d-excess records over six 400 yr time intervals around the transitions into the Holocene, GI-5.2, GI-8c, GI-18, GI-19.2, and GI-20 (Supplementary Figure 1 shows all transitions over 600 yr time intervals). Mean $\delta^{18}O$ and d-excess levels are about the same in both ice cores. Indeed, the sets of records are very similar in terms of amplitude and timing at multi-decadal-scale and show the well-known pattern comprising a cold phase, a transition, and a warm phase (characterized by less negative $\delta^{18}O$ and larger d-excess values), previously identified in Greenland ice cores[17,20]. Its simplest explanation is that atmospheric warming ($\delta^{18}O$) over the ice sheet takes place roughly in parallel to a moisture source shift to a cooler region (d-excess), although other factors contribute to d-excess changes. Indeed, the values of d-excess in Greenland precipitation are thought to be primarily an indicator of conditions over the subtropical North Atlantic[19], with possible small contributions from other source areas[43]. Although d-excess bears a signature of vapor source characteristics (sea-surface temperature (SST) and relative humidity), it is also affected by changes in condensation temperature, the temperature difference between the source and condensation regions[17,44] and possibly different changes in seasonality at the two sites[45] and by North Atlantic sea-ice removal[46]. A simple visual observation of the two records suggests that the variability is slightly smaller in NEEM compared with NGRIP. This is likely related to the stronger thinning rate in the NEEM glacial section, leading to the fact that each individual sample is the average isotopic values over a longer period at NEEM than at NGRIP. The difference is related to the influence

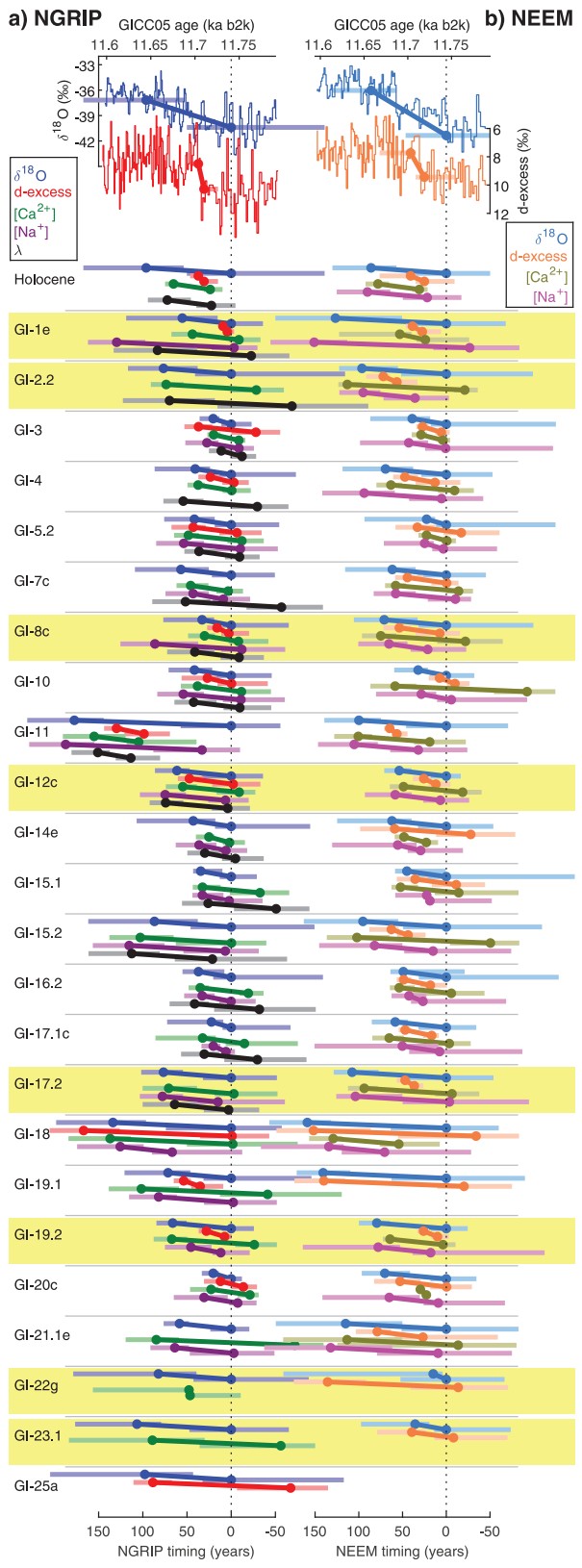

**Fig. 2 Anatomy of Last Glacial abrupt changes inferred from an ice-core multi-tracer approach.** Onset and endpoints (dots) of the studied transitions (oblique lines) towards each GI over the past 112 ka, together with associated uncertainty intervals (horizontal shaded lines) found by the ramp-fitting analysis on (**a**) NGRIP and (**b**) NEEM ice-core tracers: $\delta^{18}O$, d-excess, $[Ca^{2+}]$, $[Na^+]$ and annual-layer thickness $\lambda$ (see legend for colors). On the top, NGRIP and NEEM $\delta^{18}O$ and d-excess records are represented across the Holocene onset together with the fitted ramp to illustrate how the ramp results are represented below. Transitions preceded by stadials containing Heinrich events are indicated in yellow. All timings are shown relative to the onset of the $\delta^{18}O$ transition (dashed vertical line). The vertical amplitude between the onset and the end of each transition is the same for all tracers, it has been set arbitrarily and does not represent the true amplitude of change for each ice-core tracer.

and end) and duration of the transitions into GIs in $\delta^{18}O$, d-excess, $\lambda$, $[Na^+]$ and $[Ca^{2+}]$ relative to the onset of the $\delta^{18}O$ transition, together with the uncertainty intervals taken as the marginal posterior 5–95% credible intervals. The uncertainty intervals associated with the timing of the transition onset have a width varying from 1 to up to 205 yr depending on the considered event and ice tracer, with an average of 71 yr. First, we observe that for most studied transitions, the onsets in the four tracers occur within a few decades. In 15 out of the 21 transitions with NEEM $\delta^{18}O$, d-excess, $[Na^+]$ and $[Ca^{2+}]$ data, all onsets fall within ≤40 yr of each other. In NGRIP, five out of the seven transitions with $\delta^{18}O$, d-excess, $[Na^+]$, $[Ca^{2+}]$ and $\lambda$ data start within ≤40 yr of each other. Second, we observe large variability of the relative timing between the onsets of different tracers between events, but the observed leads and lags are consistent with zero lag considering the uncertainty intervals. Still, we observe that for eight out of 13 transitions, the lead/lag relationship between $\delta^{18}O$ and d-excess is consistent between the two cores. Also, for all events except GI-10 and GI-12c, an agreement within the uncertainty intervals is observed in the transition onsets for $\delta^{18}O$, d-excess, $[Na^+]$ and $[Ca^{2+}]$ between NEEM and NGRIP. In summary, we do not identify a clear dominant pattern in the sequence of changes of the different tracers within their uncertainty intervals when looking at the transitions individually, in line with the ramp-fitting analysis of ref. [23]. Finally, our results show that the transition for any given tracer never reaches the post-transition stable level before the other two (or three) tracers have started to change, suggesting a very tight coupling between the different climatic components reflected by the proxies.

Finally, we focus on the inferred duration of the transitions in ice tracers. The transitions in the five parameters vary from ≤10 yr to almost 200 yr, with 19 of the 24 transitions in NEEM and 20 of the 25 transitions in NGRIP being faster than a century (Fig. 4). The associated uncertainty intervals have an average width of ~117 yr, varying from 2 to 288 yr, depending on the considered event and ice tracer (Supplementary Data 4). Interestingly, we also see large variability in the transition duration from one event to the other in all tracers, with the same overall pattern observed in the two ice cores. Some transitions are 10–20 yr long in the d-excess and more gradual (~70 yr long or more) in $\delta^{18}O$, $[Ca^{2+}]$ and $[Na^+]$ (e.g., GI-2.2., GI-11, GI-17.2, and GI-19.2), such as previously highlighted across the Last Deglaciation abrupt warmings[25] (Fig. 3). A few transitions are characterized by medium-long duration (40–60 yr) in the different tracers (e.g., GI-3, GI-4, and GI-5.2). We also observe transitions characterized in both ice cores by d-excess transitions of similar or longer durations than in $\delta^{18}O$ (e.g., GI-5.2, GI-14e, and GI-18). In particular, the d-excess and $\delta^{18}O$ and transitions into GI-18 stand out as they both take ≥150 yr. The consistency in duration

on flow-induced thinning of the presence of bottom melting at NGRIP[5,38], which results in NGRIP annual layers being thicker by a factor of 1.5–2 compared with NEEM during most of the Glacial[38].

Then, we present the inferred timing of changes in ice tracers at the D-O warming onsets. Figs. 2 and 3 show the timing (onset

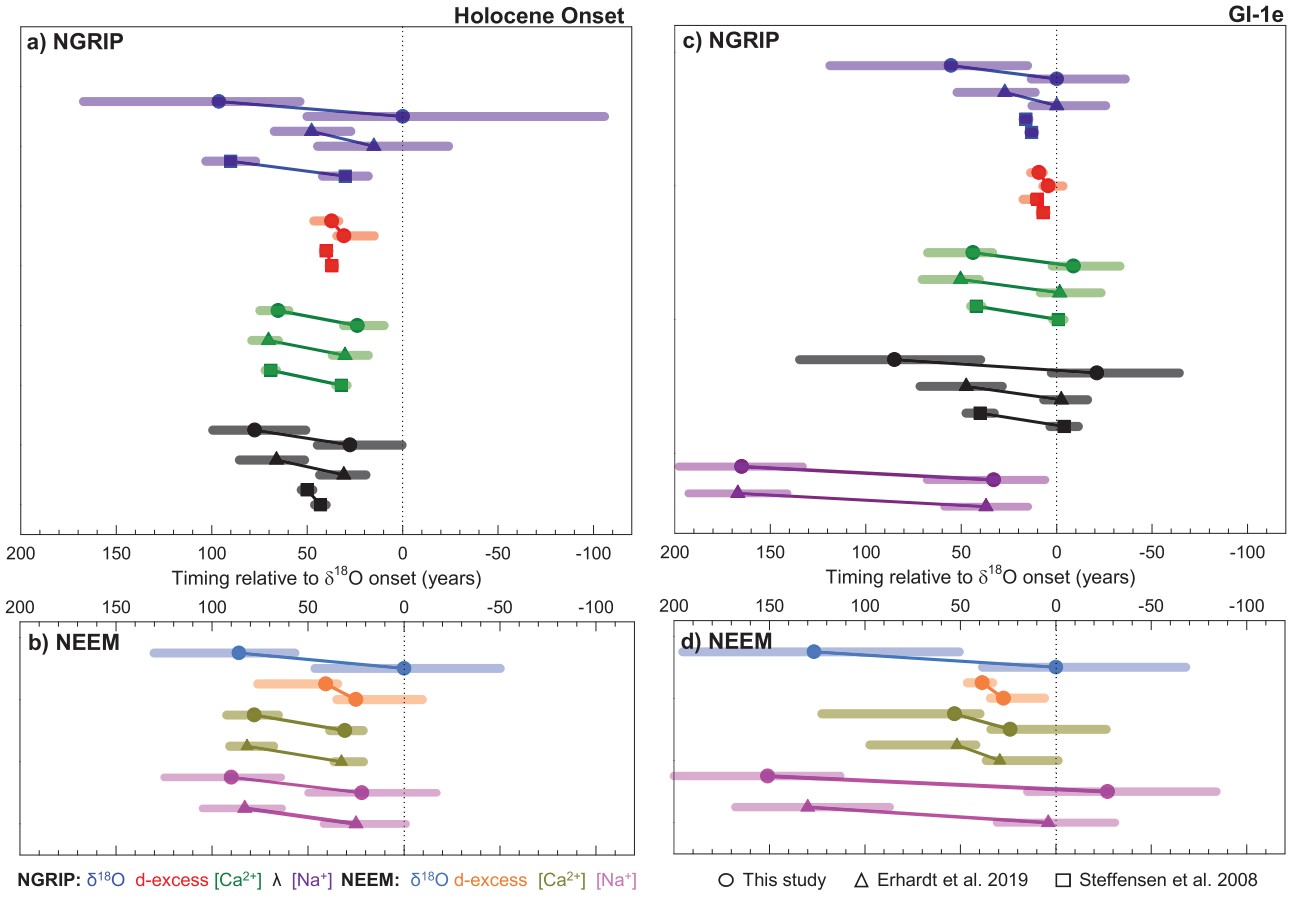

**Fig. 3 Anatomy of last deglaciation abrupt changes from an ice-core multi-tracer approach.** Onset and endpoints (symbols) of the studied transitions (oblique lines) together with associated uncertainty intervals (horizontal shaded lines) found by the ramp-fitting analyses applied to NGRIP and NEEM ice-core tracers across the Holocene onset (**a** and **b**) and across the transition into GI-1e (**c** and **d**) in this study (circles), ref. [23] (triangles) and ref. [25] (squares). All timings are represented relative to the timing of the onset in the $\delta^{18}O$ transition inferred in this study. The vertical amplitude between the onset and the end of each transition is the same for all tracers, it has been set arbitrarily and does not represent the true amplitude of change for each ice-core tracer.

patterns between the two ice cores breaks down at the bottom of the cores, i.e., in the time period earlier than GI-21.1e. We interpret this as an effect of the decreasing temporal resolution of the records through the flow-induced thinning of the ice at those depths. Because of these possible limitations, we do not consider those transitions in the following discussion.

**Discussion**

First, we discuss our results over the Last Deglaciation abrupt warmings, i.e., across the onsets of the Holocene and GI-1e (Bølling), and we compare them with results from refs. [23,25] (Fig. 3). The sequence of events obtained from NGRIP records is consistent with the previous studies within the uncertainty range and is also consistent with the sequence of events deduced from the NEEM data set. d-excess transition durations as estimated in the GRIP, GISP2, and Dye 3 ice cores were found to be systematically shorter than the $\delta^{18}O$ transitions[23] and our results on both NGRIP and NEEM ice cores suggest that overall, the transitions in $\delta^{18}O$, $\lambda$, $[Ca^{2+}]$ and $[Na^+]$ last several decades while these d-excess transitions take ≤10 yr. However, considering the uncertainty intervals of the respective transition onsets, we do not identify a statistically significant sequence of changes in NGRIP characterizing both the Holocene onset and the transition into GI-1e, i.e., different phasing is possible between tracers considering the uncertainty intervals. We provide further evidence as the NEEM results confirm that within the uncertainty intervals,

the data are consistent with both a synchronous change of $\delta^{18}O$ and $[Ca^{2+}]$ and a lead of $[Ca^{2+}]$ over $\delta^{18}O$. Hence, considering the uncertainty intervals estimated in our new study, the conclusions drawn from the original analysis of the Last Deglaciation abrupt warmings[25] that suggested a beginning of the abrupt changes in the lower latitudes should be interpreted with caution. Our observation that the onset of the two abrupt transitions occurs within a few decades of each other in the four tracers points to tight coupling and similar large-scale teleconnections between sea ice, atmospheric circulation, and isotopic temperature changes across both events. Assuming that abrupt climate change involves altered poleward energy transport by the ocean and atmosphere[13], it is unsurprising that the parameters vary so closely in phase. As originally proposed by ref. [47] and subsequently confirmed in a wide range of model studies, any perturbation to the northern high-latitude energy balance, whether by changes in sea-ice extent, altered heat transport by the AMOC or altered heat transport by the atmosphere, results in fast (annual- to decadal-scale) compensating changes in the other parameters[12,48,49].

Next, we extend our discussion to the sequence of changes over Last Glacial abrupt warmings. Our analysis identifies a range of transition durations and no systematic pattern of leads and lags between the different tracers across the GS–GI transitions. Several possible interpretations of this observation can be proposed. First, we consider whether the transitions could appear to be different owing to depositional noise and local artefacts. Differences at

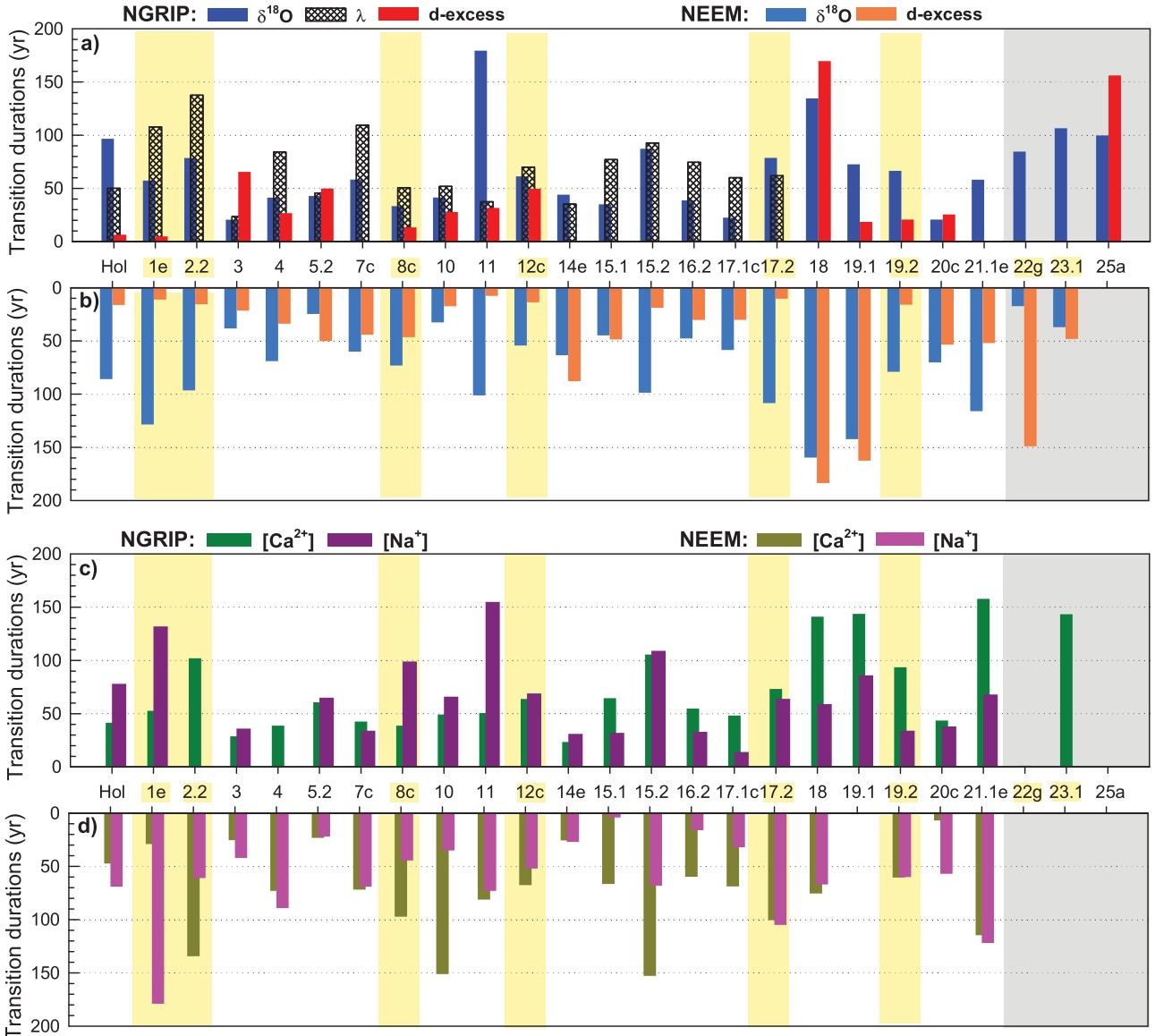

**Fig. 4 Duration estimates of the $\delta^{18}O$, d-excess, $[Ca^{2+}]$, $[Na^{2+}]$, and annual-layer thickness ($\lambda$) transitions into each GI.** Duration estimates inferred from (**a**) NGRIP data sets and (**b**) NEEM data sets. Transitions highlighted in yellow are preceded by a stadial containing a Heinrich event. Gray shading indicates the section at the bottom of the two cores where duration data should be interpreted with caution owing to marginal data resolution. Uncertainty intervals in the transition duration range from 2 to 262 yr with a mean of 86 yr (they are omitted here for clarity purposes but are shown in Supplementary Figure 4 and tabulated in Supplementary Data 4).

multi-annual scale between NGRIP and NEEM records slightly affect the relative timing of the onsets of the changes in the different tracers. The isotopic differences are likely linked to different seasonality, meso-scale atmosphere dynamics, and local processes affecting the two sites, whereas differences in the moisture source origin or transportation paths also could play a secondary role[43,45]. However, the patterns of the transition durations in NGRIP and in NEEM are overall similar despite large event-to-event variability (Fig. 1b–g), suggesting that these varying durations represent a real climate signal and are not artefacts of local depositional noise. This visual consistency (Fig. 4 and Supplementary Figure 4) is supported by testing the null hypothesis that the transition durations in the tracers and events investigated in both cores come from the same distribution (accounting for duration uncertainties, see Methods). When applied to each event individually, the null hypothesis is only rejected for two transitions out of 24 for $\delta^{18}O$, one transition out of 13 for d-excess and two transitions out of 21 for $[Ca^{2+}]$ while it

is never rejected for the studied transitions in $[Na^+]$. We also consider all the events simultaneously by testing the null hypothesis that there is no significant difference between the weighted transition durations for the two sites in each tracer. In this case, the null hypothesis is never rejected. Analyzing all transitions together and testing whether the durations correlate between the records from NGRIP and NEEM reveal significant correlation when we apply the test to (1) $\delta^{18}O$ or $[Na^+]$ records individually, (2) to the combined water isotopic tracers, (3) to the combined impurity tracers, and (4) to all tracers together (Methods, Supplementary Table 2 and Supplementary Figure 4).

Owing to the strong inter-core consistency, we argue that it is unlikely that the transitions appear different from one event to the next because of archive noise or local artefacts. Instead, we propose two alternative interpretations. The transitions could be different realizations originating from the same set of processes but expressed slightly differently between events because of

internal climate variability. Our observed timing differences between tracers at the onsets of the transitions are small, especially considering the uncertainty intervals, and internal climate variability could affect the signal propagation on regional-to-hemispheric scale and/or our ability to detect the precise onsets of transitions in different proxies. This is the underlying assumption in ref. [23] and if this assumption is fulfilled, the estimated timing differences of the individual onsets of the transitions into interstadials can be combined to reduce the influence of internal climate variability and infer the underlying archetypical sequence of events.

Alternatively, the transitions could be different from one event to the other because the mechanisms impacting abrupt changes are different. Indeed, we observe that GI-18 stands out in the two ice cores with surprisingly ~150 yr-long transitions in both $\delta^{18}O$ and d-excess. This transition occurs under atypical climatic background conditions as Marine Isotope Stage (MIS) 4 is characterized by a local ice-sheet maximum and GI-18 follows one of the longest stadials, GS-19.1, which lasts 5300 yr. Interestingly, the low-latitude counterpart of GI-18 as recorded in the Hulu cave isotopic record is not as clear as for the other abrupt events[50] and neither is it in Antarctica where the classic bipolar seesaw pattern cannot be identified unambiguously[37]. In this context, we investigate first the potential role of changes in the climatic background state (Supplementary Figure 6). However, even if the anatomy of the GI-18 transition was related to the specific climatic background state, the latter cannot explain the diversity of transition durations observed across the abrupt warming transitions within MIS 3. Here, differences are observed between neighboring events just 1–2 millennia apart, i.e., on much shorter timescales than the orbital-scale changes of the climatic background. Also, we do not find a significant correlation between the transition durations in the different ice tracers and the amplitude of key parameters of the background climate state (Supplementary Figure 6). Hence, although slow-varying forcings may influence the durations of GI and GS[51,52], they seem unable to systematically explain the differences in the durations observed between successive transitions. Finally, we investigate whether there is a link between the observed phasing and duration of transitions and the presence of large ice-rafting debris events, which occurred during some of the stadials preceding the GI onsets, the so-called Heinrich Events[53]. We find no systematic differences between the anatomy of the transitions following stadials containing Heinrich events and those without (Figs. 1–3).

Next, we provide a model perspective on the anatomy of D-O warmings. Early simulations from a coupled global ocean–atmosphere–sea-ice model forced with different freshwater amounts in the North Atlantic have shown that multiple stable or quasi-stable states of the AMOC strength could exist and that rapid transitions between these states do not have identical climate expressions as modeled temperature and precipitation changes are not linearly related to the AMOC strength[54]. Regional and local feedbacks, such as the sea-ice-margin shift and albedo feedbacks (owing to changes in sea-ice and snow extent), also come into play and complicate the response further, e.g., the relative importance of these controlling factors on the water isotopic records may differ from one event to the other leading to a variety of transitions recorded in these proxies. Owing to their design, those simulations only illustrate that transitions can be expressed differently when a single forcing (i.e., freshwater release into the North Atlantic), modulated by regional feedbacks mechanisms, is involved. However, recent modeling studies suggest that D-O events could result from unforced climate oscillations linked to dynamics internal to the climate system such as atmosphere-ice-ocean interactions altering poleward energy transport[12,13].

For comparison with our proxy-based results, we investigate the sequence of changes and the durations of the transitions in different components of the climate system across such spontaneous D-O-like oscillations in a low-resolution version of CCSM4[55]. We consider three simulations run under slightly different prescribed atmospheric $CO_2$ concentrations (185, 200, and 210 ppmv), each containing two spontaneous D-O-like transitions (Fig. 5, Supplementary Figure 7, Supplementary Table 3). Over each D-O-like transition, we extract the time series of four climatic measures from the model on the assumption that they reflect some of the same elements of the climate system as our ice-core proxy data. We look into simulated time series of (1) annual surface air temperature and (2) annual precipitation rate at the NGRIP site, (3) the sea-ice extent in the Irminger Seas, as this is the most sensitive location under interstadial-stadial changes in CCSM4 (Supplementary Figure 8), and (4) the North Atlantic Oscillation (NAO) index as a tracer for North Atlantic atmospheric circulation changes (Supplementary Figure 9, Methods). By applying our ramp-fitting-based approach to the modeled time series, we find transitions consistent with those observed in the ice-core data (Fig. 5 and Supplementary Figure 7). All modeled transitions are less than 100 yr long and can be as short as a couple of decades (the average width of the associated uncertainty intervals is 58 yr). Transitions in the simulated time series occur within a few decades of each other, with no consistent sequence within their respective uncertainty intervals. Finally, the transition in all parameters always begins before the leading parameter has reached its post-transition level. As the model climate time series do not contain archive noise, our combined model-data approach strongly supports the idea that the variability is seen in the ice tracers from one abrupt transition to the next mainly represents variability inherent to the climate system rather than noise related to how changes in the climate system are recorded in Greenland ice-core proxies. Going one step further, the model-data comparison raises the possibility that it may not be possible to resolve significant leads and lags between components of the climate system that are so tightly coupled, particularly when feedbacks between them are implicit in the nature of abrupt climate change.

The absence of a systematic pattern in the phasing or duration of transitions in different parameters, as seen in both model- and proxy-based climate parameters, could be the result of internal climate variability superimposed on a common set of mechanisms. In such a context, the ref. [23] approach of combining the estimated leads and lags for all studied transitions in order to obtain a common signal is appropriate and their resulting conclusions valid. Alternatively, there might not be a unique sequence of changes representing D-O warmings nor a unique trigger per se to these abrupt changes. These two scenarios are difficult to separate: if there are indeed multiple possible cascades of processes that can trigger a D-O event, then, the sequence may itself be sensitive to internal climate variability. The second scenario is consistent with the physics displayed in a growing number of model experiments where D-O-like oscillations can result from different processes, e.g., by forced freshwater fluxes[54], forced insolation change[56], or unforced internal oscillations[12,13]. In the case of unforced internal oscillations, destabilization of the climate system could occur owing to a range of mechanisms (e.g., stochastic atmospheric variability[57], ocean convective instability[58], and their couplings with sea-ice extent[9,13]) that could be different from one event to the next. Although the precise forcing and sequence of events vary between these examples, the end-result of abrupt climate change characterized by large sea-ice, atmospheric circulation, and temperature anomalies, is common to all.

The emerging picture of the D-O warmings is one in which the components of the climate system are so tightly coupled that it

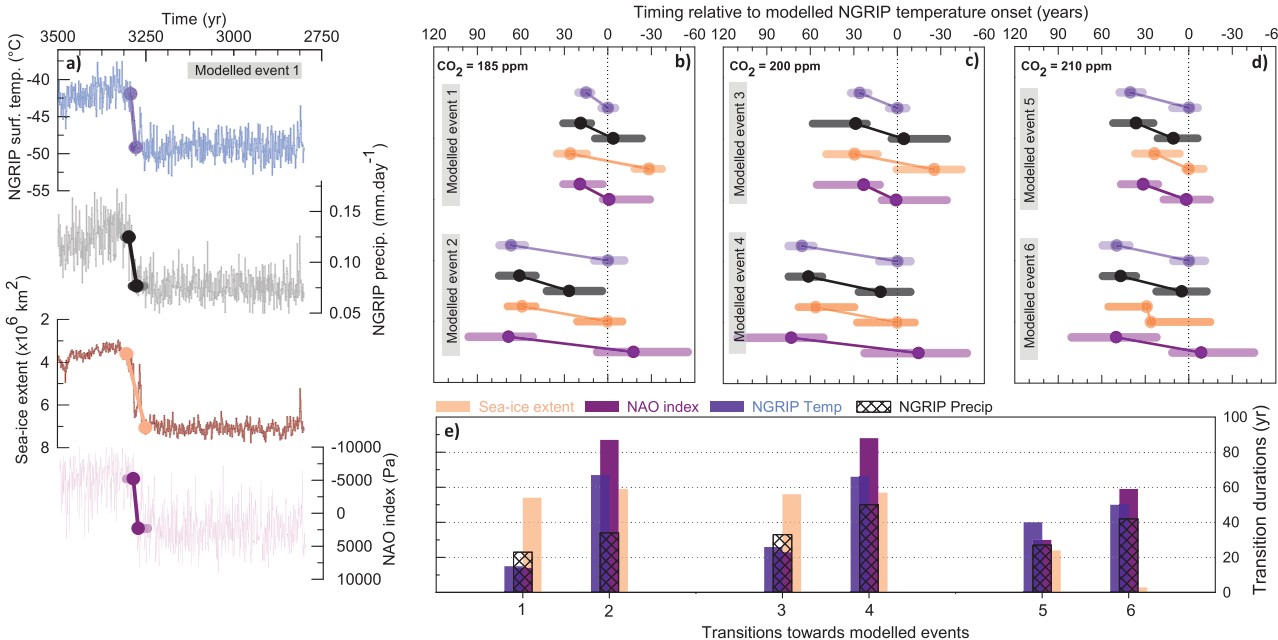

**Fig. 5 Anatomy of self-sustained abrupt transitions simulated in CCSM4.** Onset and endpoints (dots) of modeled abrupt transitions (oblique lines) together with associated uncertainty intervals (horizontal shaded lines) found by the ramp-fitting analysis on time series of the annual surface air temperature (blue) and the annual precipitation rate (black) both at the model grid point closest to NGRIP, the sea-ice extent in the Irminger Seas (light orange) and an NAO index defined as PC1 of sea-level pressure variations in the North Atlantic region (purple; details in SOM) over the two unforced oscillations simulated in CCSM4 with atmospheric $CO_2$ concentrations of (**a–b**) 185 ppm, (**c**) 200 ppm, and (**d**) 210 ppm. The time series (numbered 1–6) are shown in Supplementary Figure 7. (**a**) simulated time series for each climate parameter from the first modeled abrupt change under a $CO_2$ concentration background of 185 ppm are represented together with the resulting identification of the onset and the end of the abrupt transition from the ramp-fitting analysis to illustrate what is represented in (**b–d**). All transitions are shown relative to the timing of the onset of the NGRIP surface air temperature transition (dashed vertical line). The vertical amplitude between the onset and the end of each transition is the same for all tracers, it has been set arbitrarily and does not represent the true amplitude of change for each ice-core tracer. (**e**) Zoom on the duration estimates of the transitions in the simulated climatic parameters. Uncertainty intervals in the transition duration range from 15 to 118 yr with a mean of 57 yr (they are omitted here for clarity purposes).

may not be possible to resolve significant leads and lags between them, and consequently, it may be elusive to search for a single sequence of events in proxy data that can adequately describe all Last Glacial abrupt climate transitions. The challenge is likely to be even larger when studying the phase relationship between different types of climate archives where inter-archive dating uncertainties must also be considered. Our results underline that drawing general conclusions on the mechanisms of D-O-scale climate dynamics from phasing studies based on a single event should be avoided, as the results might not necessarily be transferred to other events. Combining our new data-based results with climate model simulations, we show that the diversity of abrupt warming transitions observed in ice-core tracers is reproduced between records and thus, not related to archive noise. Although the observed differences could be interpreted as consequences of internal climate variability superimposed onto a common set of mechanisms, the lack of a unique sequence of changes representing D-O warmings could alternatively illustrate that there is not a unique trigger per se to these abrupt changes. Such interpretation would challenge the common assumption that the rapid climatic events within the Last Glacial are the results of the same set of underlying mechanisms. Hence, although stacking of multiple abrupt transitions is useful for examining the average sequence of events and increasing the signal-to-internal-variability ratio, such an approach should be employed with caution, and the underlying assumptions in terms of physics or statistics and associated limitations should be spelled out explicitly as, for example, in ref. [23]. Finally, we support the push for (1) higher spatial coverage of well-resolved and well-dated paleoclimate records to study the range of variability in

stadials, interstadials, and the transitions between them, (2) additional Earth System Model simulations, and (3) of particular relevance to the future, more work on understanding if the glacial climate state is particularly prone to the strong sea-ice, ocean, atmosphere feedbacks that characterize the abrupt D-O transitions or if that is a general feature of the Earth's climate.

## Methods

**Greenland NGRIP and NEEM ice-core measurements.** The NEEM water isotope sections are part of the continuous high-resolution water isotope record covering 8–130 ky b2k[39]. Analyses have been performed at the Niels Bohr Institute at the University of Copenhagen using IR Cavity Ring Down Spectrometry on discrete samples with a resolution of 5 cm. The combined uncertainty of the measurements (1$\sigma$) is 0.05‰ and 0.4 ‰ for $\delta^{18}O$ and $\delta D$, respectively. In the present study, we use and show 600 yr-long sections covering 25 abrupt transitions (Supplementary Figure 1). In addition to using the existing NGRIP high-resolution $\delta^{18}O$ profile[59], we also present new d-excess data from the NGRIP ice-core for 300–500-yr-long time windows centered on 12 abrupt transitions that were measured at 5 cm resolution at the Institute of Arctic and Alpine Research (INSTAAR) Stable Isotope Lab (SIL) (University of Colorado) (Supplementary Data 1). The first set of measurements were performed in 2006–2007 across the onsets of GI-3, GI-5.2, GI-8c, GI-10, GI-11, GI-12c, GI-18, GI-19.1, GI-19.2 GI-20c, and GI-25 using an automated uranium reduction system coupled to a VG SIRA II dual-inlet mass spectrometer[60]. A second set of measurements was performed in 2016–2017 using a Picarro CRDS analyser[61] in order to fill some data gaps remaining from the 2006–2007 data set. In addition, the full section covering GI-18 as well as 20 depth levels for each other section that were already measured back in 2006, were re-measured with the Picarro instrument in order to quantify possible offsets between the old and the newer datasets. Accuracy for new NGRIP $\delta^{18}O$ and $\delta D$ measurements using the mass spectrometry-based method is 0.07‰ and 0.5‰ for $\delta^{18}O$ and $\delta D$, respectively, and is 0.1‰ and 1‰ for $\delta^{18}O$ and $\delta D$, respectively, using the laser spectroscopy-based method. The water isotope records ($\delta^{18}O$, d-excess) have a temporal resolution of better than 1 yr at 10 ka b2k, ~3 yr at 45 ka b2k, ~4 yr at 80 ka b2k, and ~5 yr at 105 ka b2k for NGRIP and of ~1 yr at 10 ka b2k, ~4 yr at 45 ka b2k, ~7 yr at 80 ka b2k, and ~18 yr at 105 ka b2k for NEEM.

The NGRIP and NEEM high-resolution $[Ca^{2+}]$ and $[Na^+]$ records over the past 60 ka are published in ref. [23], and in our study, we present the records extended back to ~108 ka b2k (Supplementary Data 2). They were measured on both ice cores using the continuous flow analysis (CFA) system of the University of Bern allowing for an annual-to-pluri-annual temporal resolution (methodological details are presented in refs. [23,62]). The effective resolution of the CFA records is between 1 and 2 cm and the relative concentration uncertainty is typically 10%. Here, we use the $[Ca^{2+}]$ and $[Na^+]$ records averaged to annual resolution. For the depth range corresponding to each year in the GICC05 time scale, average impurity concentrations are calculated directly for NGRIP. For NEEM, the corresponding NEEM depth range is first calculated by interpolation between the time scale transfer match points of ref. [40], and then the annual average impurity concentration for that depth range is calculated. The interpolation scheme thus assumes similar accumulation variability patterns between NGRIP and NEEM.

**Statistical ramp-fitting analyses of the abrupt transitions**. The assumption that the transitions are adequately described by a linear change from one stable state to another is not trivial and has been challenged previously, but neither our observations nor the current understanding of the nature of the transitions justifies employing a model with more degrees of freedom. We fit a ramp function ($f$) to the data in a window around each transition event ($Y$), the middle of which is taken from ref. [38] (Supplementary Table 1). The ramp function is parametrized by the temporal midpoint $t_{mid}$, the transition duration $\Delta t$, and the data levels before and after the transition, $y_0$ and $y_0 + \Delta y$. This formulation ensures that the ramp parameters are close to mutually independent, which is generally a desirable property for efficient probabilistic modeling. In addition to the ramp model's four parameters ($t_{mid}$, $\Delta t$, $y_0$, $\Delta y$), we model the residuals as autocorrelated noise. This introduces two additional unknown parameters for the residual variance ($\sigma^2$) and the autocorrelation length ($\tau$). These six unknown model parameters are arranged into a model vector ($m$). The likelihood ($L$) of a proposed parameter set $m$ is calculated as the probability density that the residuals ($Y-f(m)$) are a realization of red noise with the proposed noise characteristics (see ref. [23]). Credible parameter ranges are calculated from the posterior probability density obtained via Bayes' theorem:

$$P_{post}(m|Y) \propto L(Y|m) \cdot P_{prior}(m) \qquad (1)$$

where $P_{prior}$ is the prior probability density of model parameters. The prior allows us to constrain model parameters to meaningful values (see below). We draw samples from the posterior distribution using a Markov Chain Monte Carlo (MCMC) method. Here, we use an implementation of the affine invariant sampler from ref. [63] as it is highly efficient for this problem.

We use flat priors for the ramp parameters: the ramp onset and end points ($t_{mid} \pm \Delta t/2$) are constrained to the middle 80% of the data interval used for each transition, whereas $y_0$ and $\Delta y$ are constrained only by the range from minimum to maximum data values found within the interval. We use a log-normal prior for the magnitude of the additive noise ($\sigma$) centered on the standard deviation of the input data and a scale of 2. This means we assign 95% prior probability that $\sigma$ is within a factor 50 of the empirical standard deviation of $Y$. Our model estimates the autocorrelation length for each data series $\tau$ together with the ramp and noise amplitude parameters. We require that $1 \text{ yr} < \tau < 50 \text{ yr}$ with the prior distribution $p$ ($\tau$) $\propto 1/\tau$ within the interval (ref. [64], p. 161). The choice of prior on $\tau$ differs from that of ref. [23] in that it more easily allows large values of $\tau$. Initial results showed that d-excess data had no or very weak autocorrelation, which caused numerical issues for the above-mentioned implementation, so we disregarded noise autocorrelation when inferring ramp parameters for d-excess data.

The $[Ca^{2+}]$ and $[Na^+]$ data from continuous flow analysis measurements were sampled in 1 mm resolution, but annual means were used in this analysis, whereas the more sparsely sampled isotope data were used in their original measured resolution. No further resampling of the data was performed prior to analysis. The distributions of impurity data are generally closer to being log-normal than normal, and we, therefore, transform the $[Ca^{2+}]$ and $[Na^+]$ data series prior to analysis by taking the logarithm of the values.

For each data series, the MCMC sampler was run for $3\times10^6$ steps. The first 30% of the chains were discarded as burn-in, and the chains were thinned by a factor 10 which left ~ $7\times10^4$ samples from the posterior. The autocorrelation between successive samples was examined, and we generally obtained effective sample sizes in the $10^3$–$10^4$ range. The results were manually inspected for convergence, and further we verified that parameters were characterized by single maximum distributions. The median and 5–95% credible intervals (hereafter, referred to as uncertainty intervals) presented in the paper are based on the ramp kink points of all accepted models $m$, and are not derived from the distributions of $t_{mid}$ and $\Delta t$.

The ramp-fitting model was in most cases applied individually to 500-yr-long sections around the date of the GI onset provided by ref. [38] but the width of this search window could be as short as 300 yr for some events (Supplementary Table 1) as it was adjusted to account for possible data gaps or where GIs are so short that other abrupt transitions occur within the default 500-yr-long time-interval.

As a test for the robustness of our approach, we investigated how the addition of autocorrelated noise would affect the detection of the transitions. To do so, we first generated realistic noise by extracting 512 years of NEEM $\log(Ca^{2+})$ data from GS-12 (~44–43.5 ka b2k). Using the method described in refs. [65,66], a randomized noise time series with identical spectral properties (including autocorrelation) was

constructed. The noise was added to an artificial ramp of duration 50 years placed in the middle of a 512-year long data window. The ramp amplitude was 1.9 units (corresponding to a dust load decrease factor of ~7 as observed across the GI-12 onset). We note that this is a conservative approach as we are adding "stadial-strength" noise also to the interstadial part of the ramp, where the dust levels as well as the variability most often is smaller. The ramp-fitting method was run $10^6$ times for each realization of the artificial noisy ramp to produce robust statistics similar to what was done for the real transitions. The experiment was repeated 20 times with different noise resampling and the results are displayed in Supplementary Figure 2. The fitting algorithm estimates the ramp height, noise level, and autocorrelation length consistently well, and gets an average duration of 51 years. The true duration of 50 years is within the 5–95% credible intervals in 19 of the 20 cases. The ramp midpoint (not shown) was determined to be on average $0.8 \pm 4.2$ $(1\sigma)$ years later than that of the idealized noise-free ramp. The ramp duration is also not detectably influenced on average, but the autocorrelation leads to significant scatter in the results in full agreement with the analysis results from real data.

We remove from our study the results from the transitions towards GI-6, GI-9, and GI-13c as the ramp-fitting model could not provide an unequivocal solution, i.e., the inferred timing and duration of the transition were not robust within a few decades when the model was run across time windows of different widths. Supplementary Figure 3 provides an illustration of this as it displays the resulting ramps over the NGRIP $\delta^{18}O$ transitions towards GI-6 and toward GI-12c when considering three different widths of search window ($\pm250$ yr, $\pm200$ yr and $\pm100$ yr). Although the identified onsets, ends and resulting durations of the transition toward GI-12c are the same within a few years when applying the ramp-fitting model on the three different search windows (and the transition thus is included in the analysis), we observe large differences for the onset of the transition toward GI-6: the duration varies from 67 to 168 yr (and the transition is therefore excluded from further analysis). Note also that the GIC05modelext time scale older than 60 ka b2k has been constructed by combining estimates of past accumulation changes based on the $\delta^{18}O$ signal and an ice flow model to deduce the thinning of annual layers with depth. Hence, our transition analysis for layer thicknesses is limited to the most recent 60 ka.

Looking into the comparison of our results with those from ref. [25] across the last deglaciation, we observe that our uncertainty estimates are larger than those reported in the earlier study (Fig. 3). We explain this as owing to the fundamental differences in the statistical assumptions made in the RAMPFIT method[42] and ours. The approach used here explicitly models the noise superimposed on the transition including the uncertainty of the noise parameters, in turn leading to larger uncertainties in the ramp parameters.

Our new ramp-fitting model is available in Supplementary Code 1.

**Testing whether the durations of the transitions could appear to be different owing to depositional noise and local artefacts**. In order to investigate if the apparent resemblance between the durations of the transitions in NEEM and NGRIP ice records (Fig. 4) is significant, we first test for the null hypothesis that the durations, accounting for their uncertainty range, of the transitions in $\delta^{18}O$, d-excess, $[Na^+]$ and $[Ca^{2+}]$ for each event analyzed in both cores are coming from the same distribution, assuming a normal distribution for the durations of the transitions. Each tracer was considered independently from the other three. For $\delta^{18}O$, the null hypothesis is rejected for only two out of the studied 24 transitions, i.e., transitions toward GI-10 and GI-19.2. For d-excess, the null hypothesis is rejected for only one out of the studied 13 transitions, i.e., transition toward GI-19.1. For $[Ca^{2+}]$, the null hypothesis is rejected for only two out of the studied 21 transitions, i.e., transitions toward GI-10 and GI-20c. For $[Na^+]$, it is never rejected. In addition, we test the null hypothesis that there is no significant difference between the weighted transition durations for all tracers at the two sites. By testing all the events simultaneously, the null hypothesis is never rejected.

We also determine the significance of the direct correlation between NEEM transition durations and NGRIP transition durations considering seven different combinations of the data series: first, each individual tracer (1) $\delta^{18}O$ only, (2) d-excess only, (3) $[Ca^{2+}]$ only, (4) $[Na^+]$ only, then, (5) combined water isotopic tracers ($\delta^{18}O$ and d-excess), and (6) combined impurity tracers ($[Ca^{2+}]$ and $[Na^+]$), and finally, (7) all tracers together. Scatter plots are presented in Supplementary Figure 4 and they exhibit a visually clear correlation between NEEM and NGRIP transition durations even though the $R^2$ values are low (depending on the selection of data used; values not given here) because of the large scatter of the durations around the 1:1 line. We estimate the significance of the correlations for the seven different transition duration data series in the following way: we randomly permute the NEEM durations $10^5$ times (thereby, generating equally distributed uncorrelated sets of durations) and calculate the resulting slope $s$ of the (NGRIP transition duration, NEEM transition duration) plot (similar to Supplementary Figure 4) for each realization. We compute how often $|s|$ exceeds $s_0$, the slope of the unpermuted data, as a measure of the risk that a correlation of this strength is observed by chance. The results displayed in Supplementary Table 2 illustrate that except when analyzing the d-excess or $[Ca^{2+}]$ transitions durations alone, the correlations between NGRIP and NEEM transition durations are significant. In particular, when treating all transition duration pairs as one data set ($N = 77$), the significance is extremely convincing ($|s| > s_0$ for only three in $10^5$

realizations), demonstrating that despite the large scatter in durations between events and sometimes even between species, the results from the two cores show a highly consistent picture. Finally, note that comparing the results of the ramp-fitting procedure on NEEM δD and δ18O records, we observe that the timing of the onsets and the durations of the transitions are in agreement within ≤6 yr for 15 out of 19 events when considering the transition durations (Supplementary Figure 5) and 17 out of 19 events when considering the timing of the transition onset (not shown).

### Simulated unforced D-O-like oscillations

*Description of the low-resolution version of CCSM4 and of the new simulations.* We present a new set of glacial climate simulations obtained with a low-resolution version of the CCSM4 model[67]. The low-resolution version of CCSM4 is a coupled climate model of ~3.75° × 3.75° resolution in the atmosphere with 26 levels in the vertical, including a spectral dynamical core (T31). The ocean model resolution is ~3° with 60 vertical levels. The model uses POP2 and CAM4 as the ocean and atmosphere components, respectively[68–70]. The land-surface and sea-ice components are CLM4 and CICE4, respectively[71,72]. The land and atmosphere share the same Cartesian latitude and longitude grid, whereas the ocean and sea ice share the same curvilinear grid with poles centered on Greenland and Antarctica. The four earth system components are coupled using a climate coupler that conserves momentum, energy, and freshwater fluxes between earth system components (CPL7). The climate in the model has been validated against modern observations and is known to have some significant deviations from the observed climatology. Northern Hemisphere Arctic sea-ice thickness and extent is rather excessive in the model. Atlantic mass and heat transport are somewhat weak compared with the high-resolution version of the model and observations. The low heat transport results in excessive sea-ice extent and a bias in globally averaged SSTs of ~1 °C. The climate validation of the low-resolution model is fully described in ref. [55].

The model simulates glacial climate using glacial boundary conditions provided by the ICE-6G (VM5a) model for palaeotopography, bathymetry, and land ice cover[73,74]. The set-up of the boundary and initial conditions for a high-resolution glacial simulation has been discussed in detail previously[75,76]. The same format for applying the boundary and initial conditions in the high-resolution version of CCSM4 has been applied to its low-resolution version. Continental land ice is represented by Laurentide, Fennoscandian, Greenland, and Antarctic distributions representative of approximately the Last Glacial Maximum (LGM). Insolation is set according to LGM orbital parameters and an initial background atmospheric $CO_2$ concentration of 185 ppmv was used in the initial 2100 years of simulation. Two other simulations were also run to produce a set of three runs with different fixed background atmospheric $CO_2$ forcing (185, 200, and 210 ppmv) to investigate the effect of background atmospheric $CO_2$ on D-O variability. The two additional new low-resolution glacial simulations were spun off the initial 185 ppmv run at year 2100 and each of the three simulations was run for another ~5000 years. Each of the three runs exhibits two unforced climate oscillations that are approximately the same magnitude, shape and length as the D-O events observed during MIS 3, like the high-resolution simulations reported previously[12], but with periods and background variability in better agreement with observations. The branch point for each of the three sensitivity runs was chosen during the first stadial of the regular unforced D-O stadial at 185 ppmv, which was close to surface statistical equilibrium in the stadial phase of the simulation. In this study, we use the two D-O-like oscillations from each of the three simulations, totaling six different abrupt warming transitions to be analyzed using our ramp-fitting approach (Supplementary Table 3; Supplementary Figure 7). The detailed dynamics and discussion of the unforced D-O oscillations simulated in this set of low-resolution simulations will be discussed in detail elsewhere.

*Selection of modeled physical parameters to be compared with Greenland ice-core tracers.* Here, we focus specifically on the abrupt warming transitions of these spontaneous D-O-like oscillations from cold stadial to warm interstadials with the objective to look into how the signals recorded in Greenland ice cores could be reflecting the phasing of changes in the climate signals in surrounding regions. In particular, we want to investigate the existence of a variability in terms of the timing and durations of the changes in different parts of the climate system as inferred from the Greenland ice-core proxies in order to test the different explanations proposed for the observed variability in the palaeoclimate data. To do so, a set of four time series of specific climate parameters for each modeled event were extracted on the basis that they are linked to the interpretation of the ice-core tracers presented in this study. The rationales behind the choice of these parameters are discussed in the following. As the NGRIP δ18O profile is mainly controlled by the condensation temperature[77], whereas the annual-layer thickness inferred from the NGRIP GICC05 chronology[24] is determined by the annual precipitation rate and subsequent flow-induced strain, the first two modeled time series selected are the annual average surface temperature and annual precipitation changes at the model grid point closest to the NGRIP drilling site. Model studies have also highlighted that the variations in sea-ice area surrounding Greenland between GSs and GIs are also of primary importance for the temperature and precipitation response recorded in Greenland ice cores, e.g. [78,79]. In addition, sea-ice extent in the North Atlantic has been suggested to impact d-excess values as sea-ice removal during abrupt warmings could cause a change in the original

moisture source location associated with the retreating polar front in the North Atlantic[46]. In subsequent work, it has been proposed that abrupt changes in d-excess may be recording abrupt atmospheric circulation change and an associated change in moisture source location[17]. Sea ice has also a direct and significant influence on the formation and transport of sea-salt aerosol as parametrized by [Na+]. On the one hand, sea-ice formation acts as efficient, alternative source of sea-salt aerosol in addition to the open ocean. On the other hand, sea-ice extension increases the transport distance of sea-salt aerosol from the open ocean to the ice-core drill sites. [$Ca^{2+}$] changes across GS–GI transitions are also attributed to hemispheric scale atmospheric circulation patterns in addition to changes in aridity and dust storm activity in central Asian deserts[25,62]. As a result, we also look into two additional model-derived measures of the influence of the state of the atmosphere, ocean and cryosphere in the surrounding regions in the North Atlantic, i.e., a measure of annually averaged sea-ice area changes and one of the atmospheric circulation variations in the North Atlantic. In our simulations, the sea-ice extent in the North Atlantic, including the Greenland, Irminger and Norwegian (GIN) seas varies to a large degree on both seasonal and millennial timescales. One of the determining factors that drives the large variations in sea ice on longer-than-annual timescales in these simulations is the sub-sea-ice thermohaline instability that occurs prior to each D-O warming[58]. Therefore, we extract a measure of annual average sea-ice area (or concentration) in the Irminger Seas as this is the region in the model that shows the largest loss of sea ice during the warming transition (Supplementary Figure 8). Note that the model shows minor changes in the south eastern Norwegian Seas and is somewhat inconsistent with some recent observational studies[80]. This may be related to the coarse resolution of the model and the penetration of Atlantic waters into the Norwegian Seas during the stadial in this model.

As a diagnostic of the atmospheric circulation changes surrounding Greenland, we use the Hurrell PC based North Atlantic Oscillation (NAO) index[81,82] as a measure of the circulation in the North Atlantic. The method is based on a principal component analysis of the wintertime (December–January–February) changes in sea-level pressure in the region between 30 and 90°N and −90°W and 25°E. From this we can obtain both the Empirical Orthogonal Functions (EOFs) and PC time series associated with the dominant modes of variability. The EOFs in this region of the North Atlantic are well characterized by the sea-level pressure dipole pattern associated with the NAO in the modern climate (low sea-level pressure anomalies in the south east of Greenland and Iceland and high-pressure anomalies in the Azores; Supplementary Figure 9). Supplementary Figure 9 demonstrates that almost 55% of the variance (EOF1) is attributed to this general sea-level pressure anomaly associated with the NAO. The positive phase of the NAO is associated with enhanced seal-level pressure anomalies in this region (lower pressure over Iceland and higher pressure over the Azores). This in turn produces more zonal flow onto the Eurasian continent. A strong positive state in winter also increases northerly winds from Canada and Greenland, resulting in colder-than-average land and ocean temperatures in and around Greenland. The transport of dry cold air results in below-average precipitation over Greenland[83]. Through the transition, the time series of PC1 displays most of the variability associated with the transition into the interstadial. Therefore, applying the ramp-fitting procedure described in this study to the PC1 time series provides a general diagnostic for most of the circulation changes in this region and can elucidate the duration of the transition associated with atmospheric circulation changes in this sub-polar gyre region of the North Atlantic.

Looking at the ramp-fitting results, the durations of the transitions in the different modeled physical parameters seem to be slightly longer during the second transition compared with the first one (Figure 5 and Supplementary Figure 7). An explanation may lie in the length of the simulation needed to get stable statistics in a glacial climate. It is known that the AMOC may never be in a stable state and exhibit intermittency throughout an entire simulation. This is associated not only with the long timescales associated with the deep ocean to reach equilibrium but as well with natural variability associated with the ocean circulation itself. A robust characterization of the AMOC in the NCAR CESM has been shown to require long and possibly multi-millennial-scale simulations in length[84]. Therefore, the results obtained for the first set of D-O-like transitions (modeled events 1, 3, and 5, Fig. 5 and Supplementary Figure 7), might be partly affected by the fact that the model ocean may not be fully spun to equilibrium even after several thousand years. However, this potential limitation does not affect our conclusions as the variations between simulations in the second set of D-O transitions in each simulation (modeled events 2, 4, and 6), still show a clear variability amongst themselves both in term of transition duration and the leads and lags associated to the onset of the different changes and this variability is of similar order of magnitude to what is observed in the ice-core data set.

### Data availability
The new NGRIP d-excess records and the analyzed NEEM and NGRIP [$Ca^{2+}$] and [Na+] data in 1 yr resolution across a 500 yr time window covering each investigated transition are provided in the supplementary information files attached to this published article. The NGRIP high-resolution δ18O record is available at https://www.iceandclimate.nbi.ku.dk/data/ and https://doi.org/10.1594/PANGAEA.896742. The NEEM high-resolution water isotope data set is available at https://doi.org/10.1594/PANGAEA.925552. Data analyzed in this study are also available directly upon request to the authors.

## Code availability

The MATLAB script used for the ramp-fitting analysis is provided in the supplementary information files attached to this published article. It is also available directly upon request to the authors.

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

## Acknowledgements

We are very grateful to Valerie Morris for her help performing the high-resolution NGRIP water isotope analyses at INSTAAR and to Thomas Stocker for his feedback on an early version of the manuscript. This paper is an outcome of the ANA-Clim (Anatomy and Nature of Abrupt Climate Changes) project funded by the European Union's Seventh Framework Programme for research and innovation under the Marie Skłodowska-Curie grant agreement no. 600207, and the ChronoClimate project funded by the Carlsberg Foundation. This work also benefited from the financial support of the French state managed by the National Research Agency under the "Programme d'In-vestissements d'Avenir" through the Make Our Planet Great Again HOTCLIM project (ANR-19-MPGA-0001). T.E. and H.F. gratefully acknowledge the long-term support of ice-core sciences at the University of Bern by the Swiss National Science Foundation (SNF grant nos. 172506, 159563, 137635, 119612, 63333). J.B.P. received grant funding from the Australian Government. Research leading to these results benefit from the discussion as part of the ice2ice project funded by the European Research Council under the European Union's Seventh Framework Programme (FP7/2007-2013)/ERC grant agreement 610055. NGRIP was directed and organized by the Department of Geophysics at the Niels Bohr Institute for Astronomy, Physics and Geophysics, University of Copenhagen. It is supported by funding agencies in Denmark (SNF), Belgium (FNRS-CFB), France (IPEV and INSU/CNRS), Germany (AWI), Iceland (RannIs), Japan (MEXT), Sweden (SPRS), Switzerland (SNF), and the USA (NSF, Office of Polar Programs). NEEM is directed and organized by the Centre of Ice and Climate at the Niels Bohr Institute and US NSF, Office of Polar Programs. It is supported by funding agencies and institutions in Belgium (FNRS-CFB and FWO), Canada (NRCan/GSC), China (CAS), Denmark (SNF), France (IPEV, CNRS/INSU, CEA, and ANR), Germany (AWI), Iceland (RannIs), Japan (NIPR), South Korea (KOPRI), The Netherlands (NWO/ALW), Sweden (VR), Switzerland (SNF), the United Kingdom (NERC), and the USA (US NSF, Office of Polar Programs) and the EU Seventh Framework programmes Past4Future and WaterundertheIce.

## Author contributions

E.C., T.J.P., S.O.R., and J.W.C.W. designed the project. T.J.P., V.G., B.V., and B.M.V. performed the water isotopic measurements. A.G. and S.O.R. developed the ramp-fitting tool and G.V. designed, performed and analyzed the CCSM4 simulations. E.C. performed the ramp-fitting analyses on the ice-core data set, led the interpretation and discussion of the results, and wrote the manuscript with contributions from T.E., H.F., J.B.P., G.V., A. L., A.S., and S.O.R. All authors contributed to the discussion of the results and the polishing of the manuscript.

## Competing interests

The authors declare no competing interests.
