## [Peer Review File · Nature Communications]

REVIEWER COMMENTS

Reviewer #1 (Remarks to the Author):

This manuscript presents a thorough, detailed, and rigorous examination of the decade-scale changes that have come to be known as "abrupt climate change" or the Dansgaard-Oeschger events. Nearly half of the known events are interrogated, and as such this work breaks new ground in asking "how is each event different from the others?" The authors are careful in their conclusions, often writing that previously-hypothesized temporal relationships cannot be claimed on the basis of this data set due to lack of statistical significance. The conclusions of the study are modest yet important: the authors find that two possibilities remain plausible, namely 1) that internal climate-system variability (for instance, the well-known lack of global warming between 1945 and 1975) is superimposed on the mechanistic drivers of the abrupt climate change, rendering the observed climate change slightly different in each of the dozen or so studied cases; or 2) that fundamentally different mechanistic drivers may have been operative in different events.

The work is important, well-executed, clearly written, and superbly documented. This is an example of first-rate science and it will no doubt have a lasting impact. As such I believe it should be published with only very minor revision.

The one major scientific point that I believe needs clarification before this work can be published is the statement in the introduction regarding ref 23. It is written that ref 23 and earlier works (ref 21-22) showed that calcium (or dust) was reported to change about a decade prior to Greenland $\delta^{18}O$ (or temperature) at the Dansgaard-Oeschger events, and that this finding demonstrated that low-latitude dust source changes occurred prior to Greenland changes. But it is well known that wash-out by rainfall is the major mechanism removing dust from the atmosphere. Therefore these decade-leading dust results cannot be unambiguously attributed to climatic change in the dust source region. Instead, an equally probably explanation is that rainfall in the storm track belt of 40-55 deg N increased and therefore washout of dust also increased. Therefore the authors must clarify that refs 21-23 did NOT in fact demonstrate that dust source regions changed a decade prior to Greenland. Refs 21-23 can only be interpreted as finding that EITHER storm track rainfall increased OR dust source rainfall increased, or both.

The second thing I found lacking was a brief mention of the methane and $\delta^{15}N$ (temperature) phase relations found by prior work. Although these tracers are in the gas phase, and so cannot be temporally related to the ice-based tracers discussed in this manuscript, they do bear strongly on the question of whether the low latitudes or the high latitudes changed first, and as such they should be called out briefly in the introduction.

A third important point is that $\delta^{18}O$ is known to be affected by factors other than temperature. So the authors must clarify in their use of language that they do not in fact have a direct temperature proxy. For brevity, they could consider using the phrase "isotopic temperature". This phrase has minimal impact on the flow of the language, yet still adheres to scientific fact fairly well.

Minor edits:

Pg 3 line 2 It may be better to write "mid-latitude" rather than "low-latitude" given that the Taklamakan desert is at about 38 deg N, and the Taklamakan desert is the major source of Greenland ice core dust.

Pg 3 line 11 'the new NGRIP and NEEM'

Pg 23 line 23 'different datasets'

Reviewer #2 (Remarks to the Author):

Review of Capron et al.: Multiple expressions in the anatomy of past abrupt warmings recorded in Greenland ice

Capron et al. present new high-resolution ice core data of NGRIP deuterium excess (δ) and NEEM δ and $\delta^{18}O$. They combine these with previously published Ca and annual-layer thickness data to perform a multi-proxy investigation of abrupt Dansgaard-Oeschger warming transitions. They find that internal variability in the climate system and/or the proxy records is too high to make meaningful statements about the event "anatomy". They present climate model simulations of abrupt DO-like transitions that likewise do not exhibit a unique event phasing for 6 event realizations. The data fundamentally cannot distinguish whether all events exhibit the same event anatomy or all events are intrinsically different. Likewise, they cannot say whether there is a single driving mechanism for DO transitions, or multiple different ones. The manuscript clearly deserves publication in Nature communications, both on account of the extensive data sets and the important new insights. My main concern is that the authors report their findings in a way that suggests all events are different, whereas their own analysis does not appear to support this finding (at least how I understand it). This will easily confuse the casual reader – this should be rectified before publication.

The fundamental observations are that there is both a large variation in the inferred duration and timing of the transitions, as well a large uncertainty therein. The authors define a null hypothesis, namely "the transition durations in the tracers and events investigated in both cores come from the same distribution (accounting for duration uncertainties)." (P8L9). Then they show that this null hypothesis cannot be falsified at the 90% confidence level (only 2 out of 24 events violate this hypothesis). Furthermore, there is always a point in time where all tracers are in transition (In the words of the authors: "Finally, our results show that the transition for any given tracer never reaches the post-transition stable level before the other two (or three) tracers have started to change, suggesting a very tight coupling between the different climatic components reflected by the proxies.").

Given these observations, it seems surprising that the authors place so much emphasis on the fact that the transitions are all different, for example in the title, in the abstract (L18-L21) and the section header on P7L24. This emphasis seems misplaced given that the null hypothesis was not falsified. Technically their data cannot tell them whether all events are different, or the same (just subject to internal variability). This distinction is clear only upon carefully reading the paper in great detail – which I did in my capacity as reviewer. But upon casual reading I would have probably come away with a completely different conclusion, namely that they *did* falsify this hypothesis and that all transitions are fundamentally different.

Upon re-reading their null hypothesis I am not longer sure whether a single null hypothesis is tested for each event individually, or for all events simultaneously. This needs to be clarified. Ideally both tests are performed and reported.

I encourage the authors to make the language of the title, abstract and elsewhere more neutral as to whether the events are indeed different. As it stands, it seems that the authors really want this to be true, but fail to demonstrate it statistically. I personally do not care whether all events are identical (null hypothesis) or different – I simply would like to see the language used in the paper reflect the statistical reality. If I misunderstood the nature of the null hypothesis test performed, then please perform and present the correct one to provide proof for your implicit assertion that all events are not created equal.

The two hypotheses presented on P8 cannot be distinguished by the data, making the last statement of the abstract somewhat weak. Somewhat provocatively, an alternative conclusion for

the paper could be: "multi-proxy high-resolution ice core records fundamentally cannot unravel the anatomy of a DO event due to large internal climate variability". I think that may be a more powerful (and relevant?) concluding statement than currently given in the abstract. The authors hint at this when they write: "Going one step further, the model-data comparison raises the possibility that it may not be physically possible to resolve significant leads and lags between components of the climate system that are so tightly coupled, particularly when feedbacks between them are implicit in the nature of abrupt climate change." Please comment.

In the discussion, the authors should discuss two previous papers that have attempted to do the same thing as they do, namely Steffensen et al. 2008 (ref 21) and Erhardt et al. 2019 (ref 23). Given the overlap between the author teams of these three papers that may be a sensitive issue, but it is important to re-interpret those two papers in the light of the new findings. Regarding Steffensen et al. (2008), it appears that the extremely fast d transitions reported in that paper are highly anomalous in light of the full DO cycle. How do the authors interpret this? Erhardt et al. (2019) assert that there is a small (~ 1 decade) lead of Ca and layer thickness over d18O. Is that conclusion still valid with the new data?

One question I had upon reading the paper, is whether the addition of internal climate variability (or in the analysis of the authors: auto-correlated noise) would on average tend to make transitions longer, or more abrupt – intuitively I would suspect the former. Imagine the following test: take an idealized ramp with 50 years transition time; then add auto-correlated noise with the same characteristics as found in the data, and apply the fitting algorithm. If we were to do this exercise 10,000 times, what would the fitting routine find? Intuitively, I would expect that adding noise would make the transition appear longer in duration on average. This would indicate that the transition durations found in the analyses are upper bounds on the true duration of some underlying climatic parameter of interest. Please comment.

The Ca records from Erhardt (2019) are used, but are the Na data not used also? Adding it would make the paper more complete – I see no down side to doing it. Please comment.

The authors comment that it is important in the fitting routine that all parameters are independent (supplement P2 L19). However, the time scale used beyond 60ka is based on the assumption that the layer thickness scales with d18O. So the duration of the transition as given by the age scale is controlled by the d18O at that depth. In other words, Dt and Dy are not independent parameters, especially when fitting the d18O transition. Please comment.

There is not statement for data and code availability. What data are made available? I would request that the authors please also include the bag average NEEM and NGRIP d-excess and Neem bag average d18O records, which are currently unavailable online.

The model surely seems to support the first hypothesis ("The transitions could be different realizations originating from the same set of processes but expressed slightly differently between events because of internal climate variability").

Line-by-line comments:

P3L3: Asian deserts: note that this is only the case for Greenland cores

P3L26: "processes" used twice in one sentence.

P4L16: what is the mean spacing of the volcanic reference horizons?

P5L15: cooler region: or higher relative humidity. Rel humidity is the main control on deuterium excess.

P5L21: fine-scale variability is shared: can you give correlation? Can you say within dating precision?

P5L23: this explanation makes little sense to me. Do you mean a single sample averages over more time due to stronger thinning? I guess NGRIP is also a deeper site, which would also contribute to less thinning?

P6L1: are the uncertainty intervals 1sigma or 2 sigma (i.e. what is confidence interval)

P7L7: GI-1e please add: "(the Bølling transition)".

P7L10-13: This is a very selective interpretation. Because of the large uncertainty estimates you cannot exclude the possibility that they have an identical structure either.

P7L24: Section heading is again a selective interpretation. The data cannot prove that there is a non-unique sequence either. The section could just as well have been titled "No *non-unique* sequence of changes over Last Glacial abrupt warmings." Why emphasize one hypothesis and not the other?

P8L5: Can you quantify the amount of shared variability somehow? It is not obvious to me from looking at the figures alone.

P8L13: please give us the correlation coefficient. Is this the correlation between the NEEM d18O and NGRIP d18O transitions, or also between the NEEM d18O and NGRIP d transitions?

P9L3: AIM 18 is clearly visible in many Antarctic records; see for example the high-res EDC record or the WD core

P9L14: the marked events are not the classic Heinrich events. Where do they come from? These are not the events discussed by Hemming et al. (the cited ref 43). Please give the correct reference for the origin of the yellow bars.

P9L18: Perhaps not a central role, but they clearly do matter. The H-stadials are longer in duration than regular stadials, and the pattern of DO events is clearly impacted by H-events through the so-called Bond-cycle.

P9L19: again this section header is not neutral. Neither the model nor the data provide evidence for different types of DO warmings (in fact the model supports the opposite hypothesis).

P10L24: how does this compare to Erhardt who claims Ca leads (more or less the same team of authors.)

Figure 1: The choice of Heinrich stadials is very unconventional, and not supported by the cited ref. 43. Please give the correct reference, or remove.

Figure 3: could some of these data be presented as a scatter-plot? The authors claim there are correlations between several of the transitions.

Supplement:

Section 1: Why is the precision of the Picarro measurements different at both labs, despite using the same instrument? Is this a different definition? (1 sigma vs. 2 sigma?)

Which of the NGRIP events were measured with the Picarro, and which via IRMS? When were the

IRMS measurements taken?

Section 2: is the measurement noise included in the ramp fitting routine? The manuscript states that the residual is treated as autocorrelated noise, but analytical noise is not auto-correlated.

The measurement noise is considerable in d-excess (around 0.7 permil for the INSTAAR data). Maybe that explains why the d-excess residual had only weak auto-correlation (supplement P3 L13)

P6L24: condensation temperature, not site temperature

REVIEWER COMMENTS

Reviewer #1 (Remarks to the Author):

We warmly thank Reviewer 1 for her/his constructive comments that we have now accounted for in our revised version. Related changes are highlighted in green in the revised manuscript attached below our answers.

This manuscript presents a thorough, detailed, and rigorous examination of the decade-scale changes that have come to be known as "abrupt climate change" or the Dansgaard-Oeschger events. Nearly half of the known events are interrogated, and as such this work breaks new ground in asking "how is each event different from the others?" The authors are careful in their conclusions, often writing that previously-hypothesized temporal relationships cannot be claimed on the basis of this data set due to lack of statistical significance. The conclusions of the study are modest yet important: the authors find that two possibilities remain plausible, namely 1) that internal climate-system variability (for instance, the well-known lack of global warming between 1945 and 1975) is superimposed on the mechanistic drivers of the abrupt climate change, rendering the observed climate change slightly different in each of the dozen or so studied cases; or 2) that fundamentally different mechanistic drivers may have been operative in different events.

The work is important, well-executed, clearly written, and superbly documented. This is an example of first-rate science and it will no doubt have a lasting impact. As such I believe it should be published with only very minor revision.

The one major scientific point that I believe needs clarification before this work can be published is the statement in the introduction regarding ref 23. It is written that ref 23 and earlier works (ref 21-22) showed that calcium (or dust) was reported to change about a decade prior to Greenland $\delta^{18}O$ (or temperature) at the Dansgaard-Oeschger events, and that this finding demonstrated that low-latitude dust source changes occurred prior to Greenland changes. But it is well known that wash-out by rainfall is the major mechanism removing dust from the atmosphere. Therefore these decade-leading dust results cannot be unambiguously attributed to climatic change in the dust source region. Instead, an equally probable explanation is that rainfall in the storm track belt of 40-55 deg N increased and therefore washout of dust also increased. Therefore the authors must clarify that refs 21-23 did NOT in fact demonstrate that dust source regions changed a decade prior to Greenland.

Refs 21-23 can only be interpreted as finding that EITHER storm track rainfall increased OR dust source rainfall increased, or both.

> Following this comment, we have now changed the text in the revised version as follow:
Page 3, line 12: This approach was initially applied to characterise the sequence of events at the onsets of the Holocene, GI-1e (Bølling) and GI-8c^{24,25}. For each of those transitions, a lead of a few years in changes in terrestrial aerosol concentrations, accumulation rate, and mid-latitude moisture sources relative to the changes in marine aerosols and isotopic temperature was found. Such results suggest that the Greenland surface warming was preceded by changes of the conditions at the dust sources or changes to the transport to Greenland (e.g. rainfall-driven changes in aerosol washout).

The second thing I found lacking was a brief mention of the methane and d15N (temperature) phase relations found by prior work. Although these tracers are in the gas phase, and so cannot be temporally related to the ice-based tracers discussed in this manuscript, they do bear strongly on the question of whether the low latitudes or the high latitudes changed first, and as such they should be called out briefly in the introduction.

> We have now added a few sentences in the introduction mentioning the phasing between the high- and the lower-latitudes deduced from looking into $\delta^{15}\text{N}$ and CH_4 concentration records across the abrupt transitions:

Page 3, line 17: In parallel, the phasing between the high- and lower-latitude climate responses was investigated using ice-core gas phase measurements: the $\delta^{15}\text{N}$ of N_2 as a tracer for Greenland surface temperature changes^{26,27} and the methane concentration (CH_4) as a proxy for tropical climate change^{28,29}. While the first studies^{28,30} estimated a lag of a few decades of tropical CH_4 emissions behind $\delta^{15}\text{N}$ at the onset of the abrupt warmings, a more recent study³¹, focusing on the Bølling transition and using 5 yr-resolution $\delta^{15}\text{N}$ and CH_4 records, estimated that high- and low-latitude climate changes occurred essentially synchronously at that time, with Greenland surface temperature leading atmospheric methane emissions by 4.5^{+21}_{-24} yrs, in agreement within errors with ref. 24.

A third important point is that d18O is known to be affected by factors other than temperature. So the authors must clarify in their use of language that they do not in fact have a direct temperature proxy. For brevity, they could consider using the phrase "isotopic temperature". This phrase has minimal impact on the flow of the language, yet still adheres to scientific fact fairly well.

> We now refer to "isotopic temperature" where appropriate in the text and we have also written in the introduction:

Page 2, line 27: The $\delta^{18}\text{O}$ value of Greenland ice is mainly affected by local surface temperature changes, past changes in precipitation seasonality, the temperature at the moisture source regions and elevation changes¹⁴⁻¹⁷. Hence, while, $\delta^{18}\text{O}$ is not a direct temperature proxy, it can be used as a qualitative tracer of local Greenland surface temperature changes.

Minor edits:

Pg 3 line 2 It may be better to write "mid-latitude" rather than "low-latitude" given that the Taklamakan desert is at about 38 deg N, and the Taklamakan desert is the major source of Greenland ice core dust.

> Done

Pg 3 line 11 'the new NGRIP and NEEM'

> Done

Pg 23 line 23 'different datasets'

> Done

Reviewer #2 (Remarks to the Author):

We warmly thank Reviewer 2 for her/his constructive comments that we have now accounted for in our revised version. **Related changes are highlighted in yellow** in the revised manuscript attached below.

Review of Capron et al.: Multiple expressions in the anatomy of past abrupt warmings recorded in Greenland ice

Capron et al. present new high-resolution ice core data of NGRIP deuterium excess (d) and NEEM d and d18O. They combine these with previously published Ca and annual-layer thickness data to perform a multi-proxy investigation of abrupt Dansgaard-Oeschger warming transitions. They find that internal variability in the climate system and/or the proxy records is too high to make meaningful statements about the event "anatomy". They present climate model simulations of abrupt DO-like transitions that likewise do not exhibit a unique event phasing for 6 event realizations. The data fundamentally cannot distinguish whether all events exhibit the same event anatomy or all events are intrinsically different. Likewise, they cannot say whether there is a single driving mechanism for DO transitions, or multiple different ones. The manuscript clearly deserves publication in Nature communications, both on account of the extensive data sets and the important new insights. My main concern is that the authors report their findings in a way that suggests all events are different, whereas their own analysis does not appear to support this finding (at least how I understand it). This will easily confuse the casual reader – this should be rectified before publication.

The fundamental observations are that there is both a large variation in the inferred duration and timing of the transitions, as well a large uncertainty therein. The authors define a null hypothesis, namely "the transition durations in the tracers and events investigated in both cores come from the same distribution (accounting for duration uncertainties)." (P8L9). Then they show that this null hypothesis cannot be falsified at the 90% confidence level (only 2 out of 24 events violate this hypothesis). Furthermore, there is always a point in time where all tracers are in transition (In the words of the authors: "Finally, our results show that the transition for any given tracer never reaches the post-transition stable level before the other two (or three) tracers have started to change, suggesting a very tight coupling between the different climatic components reflected by the proxies.").

Given these observations, it seems surprising that the authors place so much emphasis on the fact that the transitions are all different, for example in the title, in the abstract (L18-L21) and the section header on P7L24. This emphasis seems misplaced given that the null hypothesis was not falsified. Technically their data cannot tell them whether all events are different, or the same (just subject to internal variability). This distinction is clear only upon carefully reading the paper in great detail – which I did in my capacity as reviewer. But upon casual reading I would have probably come away with a completely different conclusion, namely that they *did* falsify this hypothesis and that all transitions are fundamentally different.

Upon re-reading their null hypothesis I am not longer sure whether a single null hypothesis is tested for each event individually, or for all events simultaneously. This needs to be clarified. Ideally both tests are performed and reported.

> In the submitted manuscript, we had tested a single null hypothesis for each event individually. It is now clarified in the revised manuscript. We also now added a test where the null hypothesis is tested for all events simultaneously.

Text in the main manuscript:

Page 9, line 7: When applied to each event individually, the null hypothesis is only rejected for two transitions out of 24 for $\delta^{18}\text{O}$, one transition out of 13 for d-excess and two transitions out of 21 for $[\text{Ca}^{2+}]$ while it is never rejected for the studied transitions in $[\text{Na}^+]$. We also consider all the events simultaneously by testing the null hypothesis that there is no significant difference between the weighted transition durations for the two sites in each tracer. In this case, the null hypothesis is never rejected.

Text in the SOM:

Page 6, line 11: In addition, we test the null hypothesis that there is no significant difference between the weighted transition durations for all tracers at the two sites. By testing as such all the events simultaneously, the null hypothesis is never rejected.

I encourage the authors to make the language of the title, abstract and elsewhere more neutral as to whether the events are indeed different. As it stands, it seems that the authors really want this to be true, but fail to demonstrate it statistically. I personally do not care whether all events are identical (null hypothesis) or different – I simply would like to see the language used in the paper reflect the statistical reality. If I misunderstood the nature of the null hypothesis test performed, then please perform and present the correct one to provide proof for your implicit assertion that all events are not created equal.

> We have made a series of changes in the revised manuscript to answer the reviewer's concern and be more neutral in the text as to whether the events are indeed different:

New title: The anatomy of past abrupt warmings recorded in Greenland ice

New abstract: Data availability and temporal resolution make it challenging to unravel the anatomy – the duration and temporal phasing – of the Last Glacial abrupt climate changes. Here, we address these limitations by investigating the anatomy of abrupt changes using sub-decadal-scale records from two Greenland ice cores. We highlight the absence of a systematic pattern in the anatomy of abrupt changes as recorded in different ice parameters. This diversity in the sequence of changes seen in ice-core data is also observed in climate parameters derived from Earth System Model simulations which exhibit self-sustained abrupt variability arising from internal atmosphere-ice-ocean interactions. Our analysis of two ice cores shows that the diversity of abrupt warming transitions represents variability inherent to the climate system and not archive-specific noise. Our results hint that it may not be possible to infer statistically-robust leads and lags between the different components of the climate system because of their tight coupling.

Section header on page 8, line 23: Sequence of changes during Last Glacial abrupt warmings” instead of “no unique sequence of changes over Last Glacial abrupt warmings.

Section header on page 10, line 19: Anatomy of D-O warmings: a model perspective” instead of “Different types of D-O warmings: a model perspective.

The two hypotheses presented on P8 cannot be distinguished by the data, making the last statement of the abstract somewhat weak. Somewhat provocatively, an alternative conclusion for the paper could be: “multi-proxy high-resolution ice core records fundamentally cannot unravel the anatomy of a DO event due to large internal climate variability”. I think that may be an more powerful (and relevant?) concluding statement than currently given in the abstract. The authors hint at this when they write: “Going one step further, the model-data comparison raises the possibility that it may not be physically possible to resolve significant leads and lags between components of the climate system that are so tightly coupled, particularly when feedbacks between

them are implicit in the nature of abrupt climate change.” Please comment.

> We follow the reviewer’s suggestion and we have now rephrased the abstract in the revised version of the manuscript as such:

New abstract: Data availability and temporal resolution make it challenging to unravel the anatomy – the duration and temporal phasing – of the Last Glacial abrupt climate changes. Here, we address these limitations by investigating the anatomy of abrupt changes using sub-decadal-scale records from two Greenland ice cores. We highlight the absence of a systematic pattern in the anatomy of abrupt changes as recorded in different ice parameters. This diversity in the sequence of changes seen in ice-core data is also observed in climate parameters derived from Earth System Model simulations which exhibit self-sustained abrupt variability arising from internal atmosphere-ice-ocean interactions. Our analysis of two ice cores shows that the diversity of abrupt warming transitions represents variability inherent to the climate system and not archive-specific noise. Our results hint that it may not be possible to infer statistically-robust leads and lags between the different components of the climate system because of their tight coupling.

In the discussion, the authors should discuss two previous papers that have attempted to do the same thing as they do, namely Steffensen et al. 2008 (ref 21) and Erhardt et al. 2019 (ref 23). Given the overlap between the author teams of these three papers that may be a sensitive issue, but it is important to re-interpret those two papers in the light of the new findings. Regarding Steffensen et al. (2008), it appears that the extremely fast $\delta^{18}\text{O}$ transitions reported in that paper are highly anomalous in light of the full DO cycle. How do the authors interpret this? Erhardt et al. (2019) assert that there is a small (~ 1 decade) lead of Ca and layer thickness over $\delta^{18}\text{O}$. Is that conclusion still valid with the new data?

> We agree with the reviewer that it is important to discuss those two papers in the context of our new results and we did so already in our submitted manuscript:

Page 9, line 19: The transitions could be different realizations originating from the same set of processes but expressed slightly differently between events because of internal climate variability. Our observed timing differences between tracers at the onsets of the transitions are small, especially considering the uncertainty intervals, and internal climate variability could affect the signal propagation on regional-to-hemispheric scale and/or our ability to detect the precise onsets of transitions in different proxies. This is the underlying assumption in ref. 22 and if this assumption is fulfilled the estimated timing differences of the individual onsets of the transitions into interstadials can be combined to reduce the influence of internal climate variability and infer the underlying archetypical sequence of events.

Erhardt et al. (2019) indeed find a small (~ 1 decade) lead of Ca^{2+} and layer thickness over $\delta^{18}\text{O}$ and Na^+ . The approach in their study was designed to test for the presence of a lead in changes in North Atlantic sea-ice cover over other changes in the Earth System and was extended to other lead-lag relations. The final conclusions of the paper are fundamentally based on the assumption that the onset of all DO warming events are different realizations of the same processes and could thus be combined as explicitly stated in the paper by Erhardt et al. (2019).

The study presented here does not go against this approach but simply adds another potential interpretation of the small differences observed in terms of leads and lags between the different tracers: While the diversity may result from internal climate variability superimposed on the same set of mechanism for all the transition, it could equally reflect a diversity of potential triggers and feedback pathways making each abrupt transition unique. Hence why, we formulate some recommendations in the concluding remarks of our manuscript:

Page 13, line 2: Hence, while stacking of multiple abrupt transitions is useful for examining the average sequence of events and increase the signal-to-internal-variability ratio, such approach should be employed with caution, and the underlying assumptions in terms of physics or statistics and associated limitations should be spelled out explicitly as e.g. in ref. 22.

Regarding the reference of Erhard et al. (2019), we also modified/added the following sentences in the revised manuscript:

Page 11, line 27: The absence of a systematic pattern in the phasing or duration of transitions in different parameters, as seen in both model- and proxy-based climate parameters, could be the result of internal climate variability superimposed on a common set of mechanisms. In such a context, the ref. 22 approach of combining the estimated leads and lags for all studied transitions in order to obtain a common signal is appropriate and their resulting conclusions valid.

- Discussion results from Steffensen et al. (2008):

Page 7, line 26: Sequence of changes over the Last Deglaciation abrupt warmings. First, we discuss our results across the onsets of the Holocene and GI-1e (Bølling), and compare them to results from refs 24, 22 (Figure 3). The sequence of events obtained from NGRIP records is consistent with the previous studies within the uncertainty range and is also consistent with the sequence of events deduced from the NEEM dataset.

As we state in the manuscript, we confirm the results of Steffensen et al. (2008) that the d-excess transitions over the two abrupt warmings of the Last Deglaciation are extremely abrupt, but also find that our uncertainty estimates are larger. The uncertainty estimates are different due to different statistical assumptions used for the ramp-fitting method. We now mention this in the SOM of the revised manuscript:

Page 5, line 23: Looking into the comparison of our results with those from ref. 13 across the Last Deglaciation, we observe that the uncertainty estimates are larger than those reported in the earlier study (Figure 3). We explain this as due to the fundamental differences in the statistical assumptions made in the RAMPFIT method¹⁴ and ours. The approach used here explicitly models the noise superimposed on the transition including the uncertainty of the noise parameters in turn leading to larger uncertainties in the ramp parameters.

We do not feel that it is justified to propose that the d-transitions of Steffensen et al. (2008) are highly anomalous in the light of the last Glacial events because this is not supported by robust statistical arguments. For instance, in the NEEM ice core the d-excess transition last 16 and 11 yr for the Holocene Onset and GI-1e respectively, which is comparable with some of the durations of the d-excess transitions during the Last Glacial e.g. the transitions toward GI-11 (8 yr), GI-14 (14 yr), GI-17.2 (10 yr).

However, in the revised manuscript, we now present the figure that compares the three studies over the deglaciation in the main manuscript rather than in the SOM (this is now Figure 3), we have now added a sentence to put the interpretation proposed by Steffensen et al. (2008) in the context of our new study:

Page 8, line 6: However, considering the uncertainty intervals of the respective transition onsets, we do not identify a statistically significant sequence of changes in NGRIP characterising both the Holocene onset and the transition into GI-1e, i.e., different phasing is possible between tracers considering the uncertainty intervals. We provide further evidence as the NEEM results confirm that within the uncertainty intervals, the data are consistent with both a synchronous change of $\delta^{18}\text{O}$ and $[\text{Ca}^{2+}]$ and a lead of $[\text{Ca}^{2+}]$ over $\delta^{18}\text{O}$. Hence, considering the uncertainty intervals estimated in our new study, the conclusions drawn from the original analysis of the Last

Deglaciation abrupt warmings²⁴ that suggested a beginning of the abrupt changes in the lower latitudes should be interpreted with caution.

One question I had upon reading the paper, is whether the addition of internal climate variability (or in the analysis of the authors: auto-correlated noise) would on average tend to make transitions longer, or more abrupt – intuitively I would suspect the former. Imagine the following test: take an idealized ramp with 50 years transition time; then add auto-correlated noise with the same characteristics as found in the data, and apply the fitting algorithm. If we were to do this exercise 10,000 times, what would the fitting routine find? Intuitively, I would expect that adding noise would make the transition appear longer in duration on average. This would indicate that the transition durations found in the analyses are upper bounds on the true duration of some underlying climatic parameter of interest. Please comment.

> We made a version of the suggested test. To generate realistic noise, 512 years of NEEM $\log(\text{Ca}^{2+})$ data from GS-12 (around 44-43.5 ka b2k) were extracted. The length of the interval is chosen due to numerical convenience and does not matter. Using the method described in

Rypdal, M., Early-Warning Signals for the Onsets of Greenland Interstadials and the Younger Dryas–Preboreal Transition, J. Climate, 29 (11): 4047–4056, 2016, <https://doi.org/10.1175/JCLI-D-15-0828.1>

Boers, N. Early-warning signals for Dansgaard-Oeschger events in a high-resolution ice core record. Nat. Commun. 9, 2556, 2018, <https://doi.org/10.1038/s41467-018-04881-7>

a randomized noise time series with identical spectral properties (including autocorrelation) was constructed. The noise was added to an artificial ramp of duration 50 years placed in the middle of the data window. The ramp fitting method was run 10^6 times for each realization of the artificial noisy ramp to produce robust statistics similar to what was done for the real transitions. The experiment was repeated 20 times with different noise resampling and the results are displayed in a figure now added in the SOM of the revised paper referred to as Figure S2 (see below). The fitting algorithm estimates the ramp height, noise level, and autocorrelation length consistently well, and gets an average duration of 51 years. The true duration of 50 years is within the 5-95% credible intervals in 19 of the 20 cases. The ramp midpoint (not shown) was determined to be on average 0.8 ± 4.2 (1σ) years later than that of the idealized noise-free ramp. The ramp duration is also not detectably influenced on average, but the autocorrelation leads to significant scatter in the results in full agreement with the analysis results from real data.

We have now added a full description of this test and the associated figure in the SOM of the revised manuscript:

Page 4, line 20: As a test for the robustness of our approach, we investigated how the addition of auto-correlated noise would affect the detection of the transitions. To do so, we first generated realistic noise by extracting 512 years of NEEM $\log(\text{Ca}^{2+})$ data from GS-12 (around 44-43.5 ka b2k). Using the method described in refs 10 and 11, a randomized noise time series with identical spectral properties (including autocorrelation) was constructed. The noise was added to an artificial ramp of duration 50 years placed in the middle of a 512-year long data window. The ramp amplitude was 1.9 units (corresponding to a dust load decrease factor of ~ 7 as observed across the GI-12 onset). We note that this is a conservative approach as we are adding “stadial-strength” noise also to the interstadial part of the ramp, where the dust levels as well as the variability most often is smaller. The ramp fitting method was run 10^6 times for each realization of the artificial noisy ramp to produce robust statistics similar to what was done for the real transitions. The

experiment was repeated 20 times with different noise resampling and the results are displayed in Figure S2. The fitting algorithm estimates the ramp height, noise level, and autocorrelation length consistently well, and gets an average duration of 51 years. The true duration of 50 years is within the 5-95% credible intervals in 19 of the 20 cases. The ramp midpoint (not shown) was determined to be on average 0.8 ± 4.2 (1σ) years later than that of the idealized noise-free ramp. The ramp duration is also not detectably influenced on average, but the autocorrelation leads to significant scatter in the results in full agreement with the analysis results from real data.

Figure S2. Test of ramp fitting in the presence of autocorrelated noise. 20 different artificial noisy ramps were analysed with our ramp fitting method and the resulting transition durations for each iteration (open blue circle) are displayed together with the marginal posterior 5-95% credible intervals (vertical blue bars). The true duration of 50 years (red dashed line) is within the 5-95% credible intervals in 19 of the 20 cases.

The Ca records from Erhardt (2019) are used, but are the Na data not used also? Adding it would make the paper more complete – I see no down side to doing it. Please comment.

> We have now performed the same analyses on the Na⁺ data and hence we have made a series of changes in the revised manuscript:

Page 3, line 7: Finally, changes in Na⁺ concentrations ([Na⁺]) can be interpreted as qualitative indicators of the sea-ice cover extent in the North Atlantic at the stadial-interstadial scale²² while relative site accumulation rate changes can be estimated using reconstructions of the annual-layer thickness (denoted λ)²³.

Page 5, line 1: We also present high-resolution NGRIP and NEEM [Ca²⁺] and [Na⁺] records annually-interpolated and extended back to ~108 ka b2k (SOM).

Page 6, line 18: Timing of changes in ice tracers at the D-O warming onsets. Figures 2 and 3 show the timing (onset and end) and duration of the transitions into GIs in $\delta^{18}\text{O}$, d-excess, λ , [Na⁺] and [Ca²⁺] relative to the onset of the $\delta^{18}\text{O}$ transition, together with the uncertainty intervals taken as the marginal posterior 5%-95% credible intervals. The uncertainty intervals associated with the timing of the transition onset have a width varying from 1 to up to 205 yr depending on the considered event and ice tracer, with an average of 71 yr. First, we observe that for most studied transitions, the onsets in the four tracers occur within a few decades. In 15 out of the 21 transitions with NEEM $\delta^{18}\text{O}$, d-excess, [Na⁺] and [Ca²⁺] data, all onsets fall within ≤ 40 yr of each other. In NGRIP, five out of the seven transitions with $\delta^{18}\text{O}$, d-excess, [Na⁺], [Ca²⁺] and λ data start within ≤ 40 yr of each other. Second, we observe large variability between events of the relative timing between the onsets of different tracers, but the observed leads and lags are consistent with zero lag considering the uncertainty intervals. Still, we observe that for eight out of 13 transitions, the lead/lag relationship between $\delta^{18}\text{O}$ and d-excess is consistent between the two cores. Also, for all

events except GI-10 and GI-12c, an agreement within the uncertainty intervals is observed in the transition onsets for $\delta^{18}\text{O}$, d-excess, $[\text{Na}^+]$ and $[\text{Ca}^{2+}]$ between NEEM and NGRIP. In summary, we do not identify a clear dominant pattern in the sequence of changes of the different tracers within their uncertainty intervals when looking at the transitions individually, in line with the ramp-fitting analysis of ref. ²² (e.g., Figure S4). Finally, our results show that the transition for any given tracer never reaches the post-transition stable level before the other two (or three) tracers have started to change, suggesting a very tight coupling between the different climatic components reflected by the proxies.

Duration of the transitions in ice tracers. The transitions in the **five parameters** vary from ≤ 10 yr to almost 200 yr, with 19 of 24 in NEEM and **20 of 25** in NGRIP being faster than a century (Figure 4). The associated uncertainty intervals have an average width of **~ 117 yr, varying from 2 to 288 yr** depending on the considered event and ice tracer (Table S2). Interestingly, we also see large variability in the transition duration from one event to the other in all tracers, with the same overall pattern observed in the two ice cores. Some transitions are 10-20 yr-long in the d-excess and more gradual (~ 70 yr long or more) in $\delta^{18}\text{O}$, $[\text{Ca}^{2+}]$ and $[\text{Na}^+]$ (e.g., GI-2.2., GI-11, GI-17.2 and GI-19.2), such as previously highlighted across the Last Deglaciation abrupt warmings²⁴ (Figure 3). A few transitions are characterised by medium-long durations (40-60 yr) in the different tracers (e.g., GI-3, GI-4 and GI-5.2). We also observe transitions characterised in both ice cores by d-excess transitions of similar or longer durations than in $\delta^{18}\text{O}$ (e.g., GI-5.2, GI-14e and GI-18). In particular, the d-excess and $\delta^{18}\text{O}$ and transitions into GI-18 stand out as they both take ≥ 150 yr. The consistency in duration patterns between the two ice cores breaks down at the bottom of the cores, i.e., in the time period including and earlier than GI-21.1e. We interpret this as being due to the decreasing temporal resolution of the records through the flow-induced thinning of the ice at those depths. Because of these possible limitations, we do not consider those transitions in the discussion.

Page 8, line 3: d-excess transition durations as estimated in the GRIP, GISP2 and Dye 3 ice cores were found to be systematically shorter than the $\delta^{18}\text{O}$ transitions (ref. 23) and our results on both NGRIP and NEEM ice cores suggest that overall, the transitions in $\delta^{18}\text{O}$, λ , $[\text{Ca}^{2+}]$ and $[\text{Na}^+]$ last several decades while the d-excess transitions take ≤ 10 yr.

Page 9, line 7: When applied to each event individually, the null hypothesis is only rejected for two transitions out of 24 for $\delta^{18}\text{O}$, one transition out of 13 for d-excess and two transitions out of 21 for $[\text{Ca}^{2+}]$ **while it is never rejected for the studied transitions in $[\text{Na}^+]$.** We also consider all the events simultaneously by testing the null hypothesis that there is no significant difference between the weighted transition durations for the two sites in each tracer. In this case, the null hypothesis is never rejected.

Consequently, Figures 2, 3, 4, S1 and S6 have also been revised.

The authors comment that it is important in the fitting routine that all parameters are independent (supplement P2 L19). However, the time scale used beyond 60ka is based on the assumption that the layer thickness scales with d18O. So the duration of the transition as given by the age scale is controlled by the d18O at that depth. In other words, Dt and Dy are not independent parameters, especially when fitting the d18O transition. Please comment.

> This is correct and that is why we do not interpret the model-reconstructed layer thicknesses beyond 60 ka. Unfortunately, we do not have an independent time scale beyond 60 ka, so we have no other choice but have this limitation attached to the analysis of the transitions across this older time interval. We have now added a sentence in the revised version of the SOM to spell out clearly this limitation:

Page 5, line 9: Note that the GIC05model ext time scale older than 60 ka b2k has been constructed by combining estimates of past accumulation changes based on the $\delta^{18}\text{O}$ signal and an ice flow model to deduce the thinning of annual layers with depth. Hence, our transition analysis for layer thicknesses is limited to the most recent 60 ka.

There is not statement for data and code availability. What data are made available? I would request that the authors please also include the bag average NEEM and NGRIP d-excess and Neem bag average d18O records, which are currently unavailable online.

> We will provide the MATLAB script for the ramp fitting analysis. We will also provide the high-resolution NGRIP d-excess data as a file in the supplementary materials of the paper. Since we do not present nor use the bag NEEM and NGRIP d-excess data, we do not think that it would be reasonable to release them as part of this publication. However, the bag NEEM d-excess profile will be presented and discussed in a manuscript currently in preparation (B. Vinther, personal communication). As for the NGRIP bag d-excess record, it has been published by Landais et al. 2018 (Climate of the Past, 14, 1405-1415).

The full NEEM high resolution $\delta^{18}\text{O}$ and d-excess dataset, are now presented in a manuscript by Gkinis et al. (in preparation) and the dataset has been submitted early October 2020 to the data depository PANGAEA. We will indicate the link to the dataset when becoming available.

The analysed Ca^{2+} and Na^{+} data in 1 yr resolution across a 500 yr time window covering each investigated transition will be published alongside our present manuscript.

Gkinis, V., Vinther, Bo M., Quistgaard, T., Popp, T., Faber, A.-K., Jensen, Camilla M., Lanzky, M., Lütt, A.-M., Mandrakis, V., Ørum, Niels O., Pedersen, A.-S., Vaxevani, N., Weng, Y., Capron, E., Dahl Jensen, D., Hörhold, M., Jones, T., Jouzel, J., Landais, A., Masson-Delmotte, V., Oerter, H., Rasmussen, S. O., Steen-Larsen, H.-C., Steffensen, J.-P., Sveinbjörnsdóttir, Á. E., Vaughn, B., White, J. NEEM ice core High Resolution (0.05m) Water Isotope Ratios ($18\text{O}/16\text{O}$, $2\text{H}/1\text{H}$) covering 8-129 ky b2k, submitted to PANGAEA.

We added in the revised manuscript a short paragraph regarding data and code availability:

*Page 14, line 13: **Data and code availability:** The new high-resolution NGRIP d-excess records and the analysed NEEM and NGRIP [Ca^{2+}] and [Na^{+}] data in 1 yr resolution across a 500 yr time window covering each investigated transition are provided in the supplementary materials, as well as the MATLAB script used for the ramp fitting analysis. Note that the NEEM high-resolution water isotope dataset³⁸ is available in the PANGAEA database at <https://doi.org/xx.xxxx/PANGAEA.xxxxxx>*

The link to the dataset on PANGAEA will be updated once it will be available.

The model surely seems to support the first hypothesis ("The transitions could be different realizations originating from the same set of processes but expressed slightly differently between events because of internal climate variability").

Line-by-line comments:

P3L3: Asian deserts: note that this is only the case for Greenland cores

> We are now more specific in the revised version and we write:

Page 3, line 3: The second-order parameter d-excess ($d\text{-excess} = \delta\text{D} - 8\delta^{18}\text{O}$) is commonly interpreted as a record of past changes in evaporation conditions or shifts in mid-latitude moisture sources^{16,18,19} while Ca^{2+} concentrations ($[\text{Ca}^{2+}]$) in Greenland ice cores reflect both source strength

and transport conditions from terrestrial sources, which are mainly the mid-latitude Asian deserts^{20,21}

P3L26: "processes" used twice in one sentence.

> The sentence now reads:

Page 4, line 11: These differences can be interpreted as coming from different realizations of the same set of underlying mechanisms due to noise processes in the archive and internal variability in the climate system, or alternatively as a suggestion that one common set of mechanisms or sequence of events may not adequately describe the processes of all rapid warming transitions.

P4L16: what is the mean spacing of the volcanic reference horizons?

> The mean spacing over the volcanic reference horizons between NEEM and NGRIP is 223 yr. However, we would rather not get deeper into this in the revised manuscript. We argue that providing a mean spacing over the whole record does not make much sense since it is highly variable along the core e.g. it can be as small as a couple of years and as large as several thousands of years depending on the studied time interval.

P5L15: cooler region: or higher relative humidity. Rel humidity is the main control on deuterium excess.

> We agree that the d-excess is controlled by relative humidity at the evaporation site but other factors contribute to d-excess changes. In the revised version, we write:

Page 6, line 4: Its simplest explanation is that atmospheric warming ($\delta^{18}\text{O}$) over the ice sheet takes place roughly in parallel to a moisture source shift to a cooler region (d-excess), although other factors contribute to d-excess changes. Indeed, the values of d-excess in Greenland precipitation are thought to be primarily an indicator of conditions over the subtropical North Atlantic¹⁸, with possible small contributions from other source areas⁴³. While d-excess bears a signature of vapor source characteristics (sea-surface temperature and relative humidity), it is also affected by changes in condensation temperature, the temperature difference between the source and condensation regions^{16,44} and possibly different changes in seasonality at the two sites⁴⁵ and by North Atlantic sea-ice removal⁴⁶.

P5L21: fine-scale variability is shared: can you give correlation? Can you say within dating precision?

> This paragraph has been rephrased in the revised manuscript and this statement does not appear anymore.

*Page 5, line 25: **General characteristics of NGRIP and NEEM water isotopic records.** Figure 1 displays NGRIP and NEEM high-resolution $\delta^{18}\text{O}$ and d-excess records over six 400 yr time-intervals around the transitions into the Holocene, GI-5.2, GI-8c, GI-18, GI-19.2 and GI-20 (Figure S1 shows all transitions). Mean $\delta^{18}\text{O}$ and d-excess levels are about the same in both ice cores. Indeed, the sets of records are very similar in terms of amplitude and timing at multi-decadal scale and show the well-known pattern comprising a cold phase, a transition, and a warm phase (characterized by less negative $\delta^{18}\text{O}$ and larger d-excess values), previously identified in Greenland ice cores^{16,19}. Its simplest explanation is that atmospheric warming ($\delta^{18}\text{O}$) over the ice sheet takes place roughly in parallel to a moisture source shift to a cooler region (d-excess), although other factors contribute to d-excess changes. Indeed, the values of d-excess in Greenland precipitation are thought to be primarily an indicator of conditions over the subtropical North Atlantic¹⁸, with possible small contributions from other source areas⁴³. While d-excess bears*

a signature of vapor source characteristics (sea-surface temperature and relative humidity), it is also affected by changes in condensation temperature, the temperature difference between the source and condensation regions^{16,44} and possibly different changes in seasonality at the two sites⁴⁵ and by North Atlantic sea-ice removal⁴⁶. A simple visual observation of the two records suggests that the variability is slightly smaller in NEEM compared to NGRIP. This is likely related to the stronger thinning rate in the NEEM glacial section, leading to the fact that each individual sample is the average isotopic values over a longer period at NEEM than at NGRIP. The difference is related to the influence on flow-induced thinning of the presence of bottom melting at NGRIP^{5,38} which results in NGRIP annual layers being thicker by a factor of 1.5 to 2 compared to NEEM during most of the Glacial³⁸.

P5L23: this explanation makes little sense to me. Do you mean a single sample averages over more time due to stronger thinning? I guess NGRIP is also a deeper site, which would also contribute to less thinning?

> Yes, a single sample will average over more time as thinning increases. Rasmussen et al. (CP 9, 2713-2730, 2014) report that the increased flow-induced thinning at NEEM leads to NGRIP annual layers being thicker by a factor of 1.5 to 2 during most of the glacial. The dominating factor on thinning rate in deep glacial ice is the melting at NGRIP.

Page 6, line 13: This is likely related to the stronger thinning rate in the NEEM glacial section, leading to the fact that each individual sample is the average isotopic values over a longer period at NEEM than at NGRIP. The difference is related to the influence on flow-induced thinning of the presence of bottom melting at NGRIP^{5,38} which results in NGRIP annual layers being thicker by a factor of 1.5 to 2 compared to NEEM during most of the Glacial³⁸.

P6L1: are the uncertainty intervals 1sigma or 2 sigma (i.e. what is confidence interval)

> All uncertainty estimates are given as marginal posterior 5-95% credible intervals. We do not believe that it is suitable to include a detailed discussion of probabilistic inference theory in this manuscript, but full information can be found in Gelman et al. 2013 ("Doing Bayesian Data Analysis", CRC) and references therein. We are now more specific in the revised manuscript and we state:

Page 6, line 18: Figures 2 and 3 show the timing (onset and end) and duration of the transitions into GlIs in $\delta^{18}\text{O}$, d-excess, λ , $[\text{Na}^+]$ and $[\text{Ca}^{2+}]$ relative to the onset of the $\delta^{18}\text{O}$ transition, together with the uncertainty intervals taken as the marginal posterior 5%-95% credible intervals.

And in the SOM:

Page 16, line 1: "Table S2. Ages (yr b2k) of the onset, t_2 , and end, t_1 , and equivalent depths (m), d_2 and d_1 respectively, of the studied transitions together with their durations and associated uncertainty intervals (marginal posterior 5-95% credible intervals) found by the ramp-fitting analysis on NGRIP and NEEM ice-core tracers."

P7L7: Gl-1e please add: "(the Bølling transition)".

> Done.

P7L10-13: This is a very selective interpretation. Because of the large uncertainty estimates you cannot exclude the possibility that they have an identical structure either.

> The reviewer is correct that the uncertainties attached to the transition duration have the same magnitude as the transition duration itself. However, we still think that it is reasonable to make such observation as the pattern is observed in the two ice cores. Looking also at additional data published by Steffensen et al. (2008), we also observe that d-excess transition durations as

estimated in the GRIP, GISP2 and Dye 3 ice cores are also systematically much shorter than the $\delta^{18}\text{O}$ transitions at the onset of the two warmings of the deglaciation. We have reformulated the text in the revised manuscript:

Page 8, line 3: d-excess transition durations as estimated in the GRIP, GISP2 and Dye 3 ice cores were found to be systematically shorter than the $\delta^{18}\text{O}$ transitions (ref. 23) and our results on both NGRIP and NEEM ice cores suggest that overall, the transitions in $\delta^{18}\text{O}$, λ , $[\text{Ca}^{2+}]$ and $[\text{Na}^+]$ last several decades while the d-excess transitions take ≤ 10 yr.

P7L24: Section heading is again a selective interpretation. The data cannot prove that there is a non-unique sequence either. The section could just as well have been titled "No *non-unique* sequence of changes over Last Glacial abrupt warmings." Why emphasize one hypothesis and not the other?

> We changed the title of this paragraph in the revised version to:

Page 7, line 26: Sequence of changes over Last Glacial abrupt warmings

P8L5: Can you quantify the amount of shared variability somehow? It is not obvious to me from looking at the figures alone.

> Given the large variability in time-scale precision, there is no simple way to quantify the amount of shared variability so we preferred rephrasing the paragraph in the revised manuscript as such:

Page 8, line 26: Differences at multi-annual scale between NGRIP and NEEM records slightly affect the relative timing of the onsets of the changes in the different tracers. The isotopic differences are likely linked to different seasonality, meso-scale atmosphere dynamics, and local processes affecting the two sites while differences in the moisture source origin or transportation paths could also play a secondary role^{43,45}. However, the patterns of the transition durations in NGRIP and in NEEM are overall consistent despite large event-to-event variability (Figure 1, B-G), suggesting that these structures represent a real climate signal and are not dominated by local depositional noise. This visual consistency (Figures 4 and S4) is supported by testing the null hypothesis that the transition durations in the tracers and events investigated in both cores come from the same distribution (accounting for duration uncertainties, SOM).

P8L13: please give us the correlation coefficient. Is this the correlation between the NEEM d18O and NGRIP d18O transitions, or also between the NEEM d18O and NGRIP d transitions?

> We have now further tested the correlation between the transition durations in NEEM vs the transition durations in NGRIP. In particular, we have now looked into the significance of the direct correlation between NEEM transition durations and NGRIP transition durations considering seven different combinations of the data series: first, each individual tracer (1) $\delta^{18}\text{O}$ only, (2) d-excess only, (3) $[\text{Ca}^{2+}]$ only, (4) $[\text{Na}^+]$ only, then, (5) combined water isotopic tracers ($\delta^{18}\text{O}$ and d-excess), and (6) combined impurity tracers ($[\text{Ca}^{2+}]$ and $[\text{Na}^+]$), and finally, (7) all tracers together. Then, we have measured the significance of the correlations for the seven different transition duration data series in the following way: we randomly permute the NEEM durations 10^5 times (thereby generating equally distributed uncorrelated sets of durations) and calculate the resulting slope s of the (NGRIP transition duration, NEEM transition duration) plot (similar to Figure S4) for each realization. We computed how often $|s|$ exceeds s_0 , the slope of the unpermuted data, as a measure of the risk that a correlation of this strength is observed by chance.

We have displayed the results in a new table (Table S2). This Table S2 illustrates that, except when analysing the d-excess or $[\text{Ca}^{2+}]$ transitions durations alone, the correlations between NGRIP and NEEM transition durations are significant. In particular, when treating all transition durations pairs as one data set ($N=77$), the significance is extremely convincing ($|s|>s_0$ for only 3 in 10^5 realizations), demonstrating that despite the large scatter in durations between events and sometime even between species, the results from the two cores show a highly consistent picture.

In order to illustrate this test and also following a suggestion from the reviewer, scatter plots of the dataset are presented in Figure S4. They illustrate a visually clear correlation between NEEM and NGRIP transition durations even though the R^2 values are low (depending on the selection of data used and not given here) because of the large scatter of the durations around the 1:1 line.

In the main manuscript:

Page 9, line 12: Analysing all transitions together and testing whether the durations correlate between the records from NGRIP and NEEM reveal significant correlation when we apply the test to (1) $\delta^{18}\text{O}$ or $[\text{Na}^+]$ records individually, (2) to the combined water isotopic tracers, (3) to the combined impurity tracers, and (4) to all tracers together (SOM, Table S3 and Figure S4). Due to the strong inter-core consistency, we argue that it is unlikely that the transitions appear different from one event to the next because of archive noise or local artefacts.

In the SOM:

Page 6, line 15: We also determine the significance of the direct correlation between NEEM transition durations and NGRIP transition durations considering seven different combinations of the data series: first, each individual tracer (1) $\delta^{18}\text{O}$ only, (2) d-excess only, (3) $[\text{Ca}^{2+}]$ only, (4) $[\text{Na}^+]$ only, then, (5) combined water isotopic tracers ($\delta^{18}\text{O}$ and d-excess), and (6) combined impurity tracers ($[\text{Ca}^{2+}]$ and $[\text{Na}^+]$), and finally, (7) all tracers together. Scatter plots are presented in Figure S4 and they exhibit a visually clear correlation between NEEM and NGRIP transition durations even though the R^2 values are low (depending on the selection of data used; values not given here) because of the large scatter of the durations around the 1:1 line. We estimate the significance of the correlations for the seven different transition duration data series in the following way: we randomly permute the NEEM durations 10^5 times (thereby generating equally distributed uncorrelated sets of durations) and calculate the resulting slope s of the (NGRIP transition duration, NEEM transition duration) plot (similar to Figure S4) for each realization. We compute how often $|s|$ exceeds s_0 , the slope of the unpermuted data, as a measure of the risk that a correlation of this strength is observed by chance. The results displayed in Table S2 illustrate that for except when analysing the d-excess or $[\text{Ca}^{2+}]$ transitions durations alone, the correlations between NGRIP and NEEM transition durations are significant. In particular, when treating all transition duration pairs as one data set ($N = 77$), the significance is extremely convincing ($|s| > s_0$ for only 3 in 10^5 realizations), demonstrating that despite the large scatter in durations between events and sometime even between species, the results from the two cores show a highly consistent picture.

We also added in the SOM the Figure S4 showing the dataset plotted as scatter plots as suggested by the reviewer in one of his following comments.

Figure S4. Scatter plots of the NEEM transition duration vs NGRIP transition duration for a) $\delta^{18}\text{O}$ (blue square), b) d-excess (red triangle), c) $[\text{Ca}^{2+}]$ (green dot), d) $[\text{Na}^+]$ (purple diamond). In e) all tracers ($\delta^{18}\text{O}$ in plain blue triangle, d-excess in open blue triangle, $[\text{Ca}^{2+}]$ in plain pink circle and $[\text{Na}^+]$ in open pink circles). Marginal posterior 5-95% credible intervals are also indicated (light grey). A $x=y$ line is added in dark grey in each graph.

P9L3: AIM 18 is clearly visible in many Antarctic records; see for example the high-res EDC record or the WD core

> We agree that AIM 18 is visible in EDC and WD high resolution isotopic records. However, we would still argue that the classic bipolar seesaw pattern is not clearly observed: we do not unambiguously observe a gradual increase in $\delta^{18}\text{O}$ in Antarctica from the very start of the Greenland Stadial phase unlike for other millennial-scale events.

P9L14: the marked events are not the classic Heinrich events. Where do they come from? These are not the events discussed by Hemming et al. (the cited ref 43). Please give the correct reference for the origin of the yellow bars.

> We have now clarified the reference for the origin of the events highlighted by the yellow bars in Figure 1. They come from Guillevic et al. CP (10, 2115-2133) 2014.

P9L18: Perhaps not a central role, but they clearly do matter. The H-stadials are longer in duration than regular stadials, and the pattern of DO events is clearly impacted by H-events through the so-called Bond-cycle.

> We have removed this sentence in the revised manuscript.

P9L19: again this section header is not neutral. Neither the model nor the data provide evidence for different types of DO warmings (in fact the model supports the opposite hypothesis).

> We have changed the section header in the revised version to:

Page 10, line 19: Anatomy of D-O warmings: a model perspective

P10L24: how does this compare to Erhardt who claims Ca leads (more or less the same team of authors.)

> Following up this statement, we propose in the discussion two possible implications and one of them is: *“The absence of a systematic pattern in the phasing or duration of transitions in different parameters, as seen in both model- and proxy-based climate parameters, could be the result of internal climate variability superimposed on a common set of mechanisms”*.

Based on that, it is reasonable to follow the Erhardt et al. (2019) approach and to combine the estimated leads and lags for all studied transitions in order to obtain a common signal suggesting that Ca²⁺ change is leading the changes in the other ice tracers.

In the revised manuscript, we have now added the following sentence:

Page 12, line 1: In such a context, the ref. 22 approach of combining the estimated leads and lags for all studied transitions in order to obtain a common signal is appropriate and their resulting conclusions valid.

Figure 1: The choice of Heinrich stadials is very unconventional, and not supported by the cited ref. 43. Please give the correct reference, or remove.

> We have now clarified the reference for the origin of the events highlighted by the yellow bars in Figure 1. They come from Guillevic et al. (CP, 10, 2115-2133, 2014).

Figure 3: could some of these data be presented as a scatter-plot? The authors claim there are correlations between several of the transitions.

> That is a good idea and as previously shown, we have now added a figure in the SOM of the revised manuscript presenting the data with scatter-plots (Figure S4).

Supplement:

Section 1: Why is the precision of the Picarro measurements different at both labs, despite using the same instrument? Is this a different definition? (1 sigma vs. 2 sigma?)

> The uncertainties in the submitted manuscript for the measurements performed at INSTAAR were very conservative. After recalculation, the uncertainties attached to the PICARRO measurement performed at INSTAAR are 0.1‰ and 0.4‰ for δ¹⁸O and δD respectively. We have changed the estimates accordingly in the revised manuscript.

Which of the NGRIP events were measured with the Picarro, and which via IRMS? When were the IRMS measurements taken?

> Isotopic measurements were performed with the IRMS in 2006-2007. They covered time intervals across the onsets of the Holocene, GI-1e, GI-3, GI-4, GI-5.2, GI-8c, GI-10, GI-11, GI-18, GI-20c and GI-25. Additional measurements on time intervals covering GI-3, GI-4, GI-5.2, GI-8, GI-10, GI-11 and GI-20c were performed with the PICARRO instrument in 2016-2017 in order to fill in some data gaps remaining from the 2006-2007 dataset. The full time interval covering GI-18 as well as 20 depth levels for each other interval that were already measured back in 2006-2006 were re-measured with the Picarro instrument in order to quantify possible offsets between the old and the newer datasets.

We have now added this information in the SOM of the revised manuscript:

Page 2, line 2: A first set of measurements were performed in 2006-2007 across the onsets of GI-3, GI-4, GI-5.2, GI-8c, GI-10, GI-11, GI-12c, GI-18, GI-19.1, GI-19.2 GI-20c and GI-25 via an automated uranium reduction system coupled to a VG SIRA II dual-inlet mass spectrometer³. A second set of measurements was performed in 2016-2017 using a Picarro CRDS analyser⁴ in order to fill some data gaps remaining from the 2006-2007 dataset. In addition, the full section covering GI-18 as well as 20 depth levels for each other section that were already measured back in 2006-2006 were re-measured with the Picarro instrument in order to quantify possible offsets between the old and the newer datasets.

Section 2: is the measurement noise included in the ramp fitting routine? The manuscript states that the residual is treated as autocorrelated noise, but analytical noise is not auto-correlated. The measurement noise is considerable in d-excess (around 0.7 permil for the INSTAAR data). Maybe that explains why the d-excess residual had only weak auto-correlation (supplement P3 L13)

> Our ramp fitting analysis method initially had both autocorrelated and non-correlated noise. For all other species than d-excess, the autocorrelated noise was dominant, and the model was not able to reliably estimate the magnitude of the non-autocorrelated noise contribution, which we therefore ignore. For d-excess, we had the opposite experience: due to the relatively high analytical noise levels, the model could not determine the magnitude of autocorrelated noise, which we therefore removed from the model.

P6L24: condensation temperature, not site temperature

> The sentence in the revised manuscript now reads as follows:

[revised manuscript text omitted]

NEEM high-resolution dataset³⁸ is available in the PANGAEA database at
<https://doi.org/xx.xxxx/PANGAEA.xxxxxx>

**Competing Interest statement**

The authors declare no competing financial interests.

**References**

- Voelker, A. H. L. Global distribution of centennial-scale records for Marine Isotope Stage (MIS) 3:
a database *Quaternary Science Reviews* **21**, 1185–1212 (2002).
- Capron, E. *et al.* Synchronising EDML and NorthGRIP ice cores using d¹⁸O of atmospheric oxygen
(d¹⁸O_{atm}) and CH₄ measurements over MIS 5 (80–123 kyr). *Quaternary Science Reviews* **29**, 222-
234, doi:10.1016/j.quascirev.2009.07.014 (2010).

EPICA community members. One-to-one coupling of glacial climate variability in Greenland and
Antarctica. *Nature* **444**, 195-198, doi:Doi 10.1038/Nature05301 (2006).

Johnsen, S. J. *et al.* Oxygen isotope and palaeotemperature records from six Greenland ice-core
stations: Camp Century, Dye-3, GRIP, GISP2, Renland and NorthGRIP. *Journal of Quaternary
Science* **16**, 299-307 (2001).

NorthGRIP community members. High-resolution record of Northern Hemisphere climate
extending into the last interglacial period. *Nature* **431**, 147-151 (2004).

Broecker, W. S., Peteet, D. M. & Rind, D. Does the ocean–atmosphere system have more than
one stable mode of operation? . *Nature* **315**, 21-26 (1985).

Ganopolski, A. & Rahmstorf, S. Rapid changes of glacial climate simulated in a coupled climate
model. *Nature* **409**, 153-158 (2001).

Kindler, P. *et al.* Temperature reconstruction from 10 to 120 kyr b2k from the NGRIP ice core.
*Climate of the Past* **10**, 887-902 (2014).

Dokken, T. M., Nisancioglu, K. H., Li, C., Battisti, D. S. & Kissel, C. Dansgaard-Oeschger cycles:
Interactions between ocean and sea ice intrinsic to the Nordic seas. *Paleoceanography* **28**, 491–
502, doi:410.1002/palo.20042. (2013).

Peterson, L. C., Haug, G. H., Hughen, K. A. & Röhl, U. Rapid Changes in the Hydrologic Cycle of
the Tropical Atlantic During the Last Glacial *Science* **290**, 1947-1951 (2000).

Petersen, S. V., Schrag, D. P. & Clark, P. U. A new mechanism for Dansgaard-Oeschger cycles.
*Paleoceanography* **28**, 24-30 (2013).

Vettoretti, G. & Peltier, W. R. Fast Physics and Slow Physics in the Nonlinear Dansgaard–Oeschger
Relaxation Oscillation. *Journal of Climate* <https://doi.org/10.1175/JCLI-D-17-0559.1> (2018).

Li, C. & Born, A. Coupled atmosphere-ice-ocean dynamics in Dansgaard-Oeschger events.
*Quaternary Science Reviews* **203**, 1-20 (2019).

Dansgaard, W. *et al.* North Atlantic climatic oscillations revealed by deep Greenland ice cores.
*Climate Processes and Climate Sensitivity edited by J. E. Hansen and T. Takahashi, pp. 288 –
298, AGU, Washington, D. C.* **29** (1984).

Krinner, G., Genthon, C. & Jouzel, J. GCM analysis of local influences on ice core d signals.
*Geophysical Research Letters* **24**, 2825-2828 (1997).

Masson-Delmotte, V. *et al.* Deuterium excess reveals millennial and orbital scale fluctuations of
Greenland moisture origin. *Science* **309**, 118-121 (2005).

NEEM community members. Eemian interglacial reconstructed from a Greenland folded ice core.
*Nature* **493**, 489-493, doi:410.1038/nature11789 (2013).

Johnsen, S. J., Dansgaard, W. & White, J. W. C. The origin of Arctic precipitation under present
and glacial conditions. *Tellus* **41**, 452-469 (1989).

- Landais, A. *et al.* Ice core evidence for decoupling between mid-latitude atmospheric water cycle
and Greenland temperature during the last deglaciation. *Climate of the Past* **14**, 1405-1415,
doi:10.5194/cp-2018-65 (2018).
- Markle, B. R., Steig, E. J., Roe, G. H., Winckler, G. & McConnell, J. R. Concomitant variability in
high-latitude aerosols, water isotopes and the hydrologic cycle. *Nature Geoscience*,
doi:10.1038/s41561-018-0210-9 (2018).
- Ruth, U. *et al.* Ice core evidence for a very tight link between North Atlantic and east Asian glacial
climate. *Geophysical Research Letters* **34**, L03706, <https://doi.org/10.1029/2006GL027876>
(2007).
- Erhardt, T. *et al.* Decadal-scale progression of the onset of Dansgaard–Oeschger warming events.
*Climate of the Past* **15**, 811–825 (2019).
- Svensson, A. *et al.* A 60 000 year Greenland stratigraphic ice core chronology. *Climate of the Past*
**4**, 47-57 (2008).
- Steffensen, J. P. *et al.* High-Resolution Greenland Ice Core Data Show Abrupt Climate Change
Happens in Few Years. *Science* **321**, 680-683 (2008).
- Thomas, E. R. *et al.* Anatomy of a Dansgaard-Oeschger warming transition: High-resolution
analysis of the North Greenland Ice Core Project ice core. *Journal of Geophysical Research* **114**,
**D08102**, doi:10.1029/2008JD011215. (2007).
- Landais, A. *et al.* Quantification of rapid temperature change during DO event 12 and phasing with
methane inferred from air isotopic measurements. *Earth and Planetary Science Letters* **225**, 221-
232, doi:DOI 10.1016/j.espl.2004.06.009 (2004).
- Severinghaus, J. P. & Brook, E. J. Abrupt climate change at the end of the last glacial period
inferred from trapped air in polar ice. *Science* **286**, 930-934 (1999).
- Baumgartner, M. High-resolution inter-polar difference of atmospheric methane around the Last
Glacial Maximum. . *Biogeosciences* **9**, 3961-3977 (2012).
- Brook, E. J., Harder, S., Severinghaus, J., Steig, E. J. & Sucher, C. M. On the origin and timing of
rapid changes in atmospheric methane during the last glacial period. . *Global Biogeochemical Cycle*
**14**, 559-572 (2000).
- Huber, C. *et al.* Isotope calibrated Greenland temperature record over Marine Isotope Stage 3 and
its relation to CH₄. *Earth and Planetary Science Letters* **243**, 504-519, doi:DOI
10.1016/j.epsl.2006.01.002 (2006).
- Rosen, J. L. *et al.* An ice core record of near-synchronous global climate changes at the Bølling
transition. *Nature Geoscience* DOI: 10.1038/NGEO2147 (2014).
- Markle, B. R. *et al.* Global atmospheric teleconnections during Dansgaard–Oeschger events.
*Nature Geoscience* **10**, 36–40 (2016).
- Pedro, J. B. *et al.* The last deglaciation: timing the bipolar seesaw *Climate of the Past* **7**, 671–683
(2011).

- Svensson, A. *et al.* Bipolar volcanic synchronization of abrupt climate change in Greenland and
Antarctic ice cores during the last glacial period. *Clim. Past* **16**, 1565-1580, doi:10.5194/cp-16-
1565-2020 (2020).
- WAIS Divide Project Members. Precise inter-polar phasing of abrupt climate change during the last
ice age. *Nature* doi: **10.1038/nature14401**. (2015).
- Capron, E. *et al.* Millennial and sub-millennial scale climatic variations recorded in polar ice cores
over the last glacial period. *Clim. Past* **6**, 345-365, doi:10.5194/cp-6-345-2010 (2010).
- Rasmussen, S. O. *et al.* A stratigraphic framework for abrupt climatic changes during the Last
Glacial period based on three synchronized Greenland ice-core records: refining and extending the
INTIMATE event stratigraphy. *Quaternary Science Reviews* **106**, 14-28,
doi:<http://dx.doi.org/10.1016/j.quascirev.2014.09.007> (2014).
- Gkinis, V. *et al.* NEEM ice core High Resolution (0.05m) Water Isotope Ratios ($^{18}\text{O}/^{16}\text{O}$, $^2\text{H}/^1\text{H}$)
covering 8-129 ky b2k. *submitted to PANGAEA*.
- Svensson, A. *et al.* A 60 000 year Greenland stratigraphic ice core chronology. *Clim. Past* **4**, 47-
57, doi:10.5194/cp-4-47-2008 (2008).
- Rasmussen, S. O. *et al.* A first chronology for the North Greenland Eemian Ice Drilling (NEEM) ice
core. *Climate of the Past* **9**, 2713–2730, <https://doi.org/2710.5194/cp-2719-2713-2013> (2013).
- Wolff, E., Chappellaz, J., Blunier, T., Rasmussen, S. O. & Svensson, A. Millennial-scale variability
during the last glacial: The ice core record. *Quaternary Science Reviews* **29** (2010).
- Mudelsee, M. Ramp function regression: a tool for quantifying climate transitions. *Computers &*
*Geosciences* **26**, 293-307 (2000).
- Charles, C., Rind, D., Jouzel, J., Koster, R. & Fairbanks, R. Glacial–interglacial changes in moisture
sources for Greenland: influences on the ice core record of climate. *Science* **261**, 508-511 (1994).
- Steen-Larsen, H. C. *et al.* Continuous monitoring of summer surface water vapor isotopic
composition above the Greenland ice sheet. *Atmos. Chem. Phys.* **13**, 4815–4828 (2013).
- Guillevic, M. *et al.* Spatial gradients of temperature, accumulation and $\delta^{18}\text{O}$ -ice in Greenland over
a series of Dansgaard–Oeschger events. *Climate of the Past* **9**, 1029-1051 (2013).
- Dansgaard, W., White, J. W. C. & Johnsen, S. J. The abrupt termination of the Younger Dryas
climate event. *Nature* **339** (6225), 532-534 (1989).
- Bjerknes, J. Atlantic air-sea interaction. *Advances in Geophysics* **10**, 1-82 (1964).
- Outten, S., Esau, I. & Otterå, O. H. Bjerknes Compensation in the CMIP5 Climate Models. *Journal*
*of Climate* **31**, 8745-8760 (2018).
- Tomas, R. A., Deser, C. & Sun, L. The role of ocean heat transport in the global climate response
to projected Arctic sea ice loss. *Journal of Climate* **29**, 6841–6859 (2016).
- Wang, Y. J. *et al.* A High-Resolution Absolute-Dated Late Pleistocene Monsoon Record from Hulu
Cave, China. *Science* **94**, 2345-2348 (2001).

- Boers, N., Ghil, M. & Rousseau, D. D. Ocean circulation, ice shelf, and sea ice interactions explain
Dansgaard–Oeschger cycles *Proceedings of the National Academy of Sciences* **115** (47) E11005–
E11014, <https://doi.org/10.1073/pnas.1802573115> (2018).
- Lohmann, J. & Ditlevsen, P. Objective extraction and analysis of statistical features of Dansgaard–
Oeschger events. *Climate of the Past* **15**, 1771–1792 (2018).
- Hemming, S. R. Heinrich events: massive Late Pleistocene detritus layers of the North Atlantic and
their global climate imprint, . *Reviews of Geophysics* **4**, RG1005, doi:10.1029/2003RG000128
(2004).
- Flückiger, J., Knutti, R., White, J. W. C. & Renssen, H. Modeled seasonality of glacial abrupt climate
events. *Climate Dynamics* **31**, 633–645, doi:10.1007/s00382-008-0373-y (2008).
- Shields, C. A. *et al.* The low-resolution CCSM4. *Journal of Climate*, [https://doi.org/10.1175/JCLI-](https://doi.org/10.1175/JCLI-D-1111-00260.00261)
[D-1111-00260.00261](https://doi.org/10.1175/JCLI-D-1111-00260.00261) (2012).
- Nielsen, S. B., Jochum, M., Pedro, J. B., Eden, C. & Nuterman, R. Two-Timescale Carbon Cycle
Response to an AMOC Collapse. *Paleoceanography and Paleoclimatology*,
<https://doi.org/10.1029/2018PA003481> (2019).
- Kleppin, H., Jochum, M., Otto-Bliesner, B., Shields, C. A. & Yeager, S. Stochastic Atmospheric
Forcing as a Cause of Greenland Climate Transitions. *Journal of Climate* **28**, 7741–7763 (2015).
- Vettoretti, G. & Peltier, W. R. Thermohaline instability and the formation of glacial North Atlantic
super polynyas at the onset of Dansgaard-Oeschger warming events. *Geophys. Res. Letters*,
10.1002/2016GL068891 (2016).
- Guillevic, M. *et al.* Evidence for a three-phase sequence during Heinrich Stadial 4 using a
multiproxy approach based on Greenland ice core records. *Climate of the Past* **10** (2014).

**Figure 1. Abrupt climate variability recorded in Greenland water isotopic records.**

(a) NGRIP $\delta^{18}\text{O}$ record⁵. Studied abrupt warming transitions are highlighted with red vertical bars and
Greenland Interstadials (GI) are numbered³⁷. Grey boxes indicate intervals shown in panels (b) to (g)
illustrating the variety of abrupt GS-GI transitions across the Last Glacial; Stadials containing Heinrich
events are indicated in yellow following refs 53, 59, and Marine Isotope Stages (MIS) are indicated in
grey. (b-g) High-resolution $\delta^{18}\text{O}$ from NGRIP (dark blue) and NEEM (light blue) and d-excess from NGRIP
(red) and NEEM (orange) over 400 yr time intervals centered on the Holocene abrupt onset (b) and the
abrupt transitions into GI-5.2 (c), GI-8c (d), GI-18 (e), GI-19.2 (f) and GI-20c (g).

**Figure 2. Anatomy of Last Glacial abrupt changes inferred from an ice-core multi-tracer approach.**

Onset and end points (dots) of the studied transitions (oblique lines) towards each GI over the past 112
11 ka, together with associated uncertainty intervals (horizontal shaded lines) found by the ramp-fitting
analysis on (a) NGRIP and (b) NEEM ice-core tracers: $\delta^{18}\text{O}$, d-excess, $[\text{Ca}^{2+}]$, $[\text{Na}^+]$ and annual layer
thickness λ (see legend for colours). On the top, NGRIP and NEEM $\delta^{18}\text{O}$ and d-excess records are
represented across the Holocene onset together with the fitted ramp to illustrate how the ramp results are
represented below. Transitions preceded by stadials containing Heinrich events are indicated in yellow.
All timings are shown relative to the onset of the $\delta^{18}\text{O}$ transition (dashed vertical line). The vertical
amplitude between the onset and the end of each transition is the same for all tracers, it has been set
arbitrarily and does not represent the true amplitude of change for each ice core tracer.

**Figure 3. Anatomy of Last Deglaciation abrupt changes from an ice-core multi-tracer approach.**

Onset and end points (symbols) of the studied transitions (oblique lines) together with associated
uncertainty intervals (horizontal shaded lines) found by the ramp-fitting analyses applied to NGRIP and
NEEM ice core tracers across the Holocene onset (a and b) and across the transition into GI-1e (c and d)
in this study (circles), ref. 22 (triangles) and ref. 24 (squares). All timings are represented relative to the
timing of the onset in the $\delta^{18}\text{O}$ transition inferred in this study. The vertical amplitude between the onset
and the end of each transition is the same for all tracers, it has been set arbitrarily and does not represent
the true amplitude of change for each ice core tracer.

**Figure 4. Duration estimates of the $\delta^{18}\text{O}$, d-excess, $[\text{Ca}^{2+}]$, $[\text{Na}^{2+}]$ and annual layer thickness (λ)**
**transitions into each GI.**

Duration estimates inferred from (a) NGRIP datasets and (b) NEEM datasets. Transitions highlighted in
yellow are preceded by a stadial containing a Heinrich event. Grey shading indicates the section at the
bottom of the two cores where duration data should be interpreted with caution due to marginal data
resolution. Uncertainty intervals in the transition duration range from 2 to 262 yr with a mean of 86 yr
(they are omitted here for clarity purposes but are shown in Figure S4 and tabulated in Table S2).

**Figure 5. Anatomy of self-sustained abrupt transitions simulated in CCSM4.**

Onset and end points (dots) of modeled abrupt transitions (oblique lines) together with associated
uncertainty intervals (horizontal shaded lines) found by the ramp-fitting analysis on time series of the
annual surface air temperature (blue) and the annual precipitation rate (black) both at the model grid point
closest to NGRIP, the sea-ice extent in the Irminger Seas (light orange) and a NAO index defined as PC1
of sea level pressure variations in the North Atlantic region (purple; details in SOM) over the two unforced
oscillations simulated in CCSM4 with atmospheric CO_2 concentrations of (b) 185 ppm, (c) 200 ppm and
(d) 210 ppm. The time series (numbered 1-6) are shown in Figure S7. In (a), simulated time series for
each climate parameter from the first modeled abrupt change under a CO_2 concentration background of
185 ppm are represented together with the resulting identification of the onset and the end of the abrupt
transition from the ramp-fitting analysis to illustrate what is represented in b-d. All transitions are shown
relative to the timing of the onset of the NGRIP surface air temperature transition (dashed vertical line).
The vertical amplitude between the onset and the end of each transition is the same for all tracers, it has
been set arbitrarily and does not represent the true amplitude of change for each ice core tracer.

(e) Zoom on the duration estimates of the transitions in the simulated climatic parameters. Uncertainty
intervals in the transition duration range from 15 to 118 yr with mean of 57 yr (they are omitted here for
clarity purposes).

1 Figure 1.

2

3

1 Figure 2.

2

3

1 Figure 3.

2

3

1 **Figure 4.**

2

3

4

1 **Figure 5**

2

**The anatomy of past abrupt warmings recorded in Greenland ice**

E. Capron^{1,2#}, S. O. Rasmussen¹, T. J. Popp¹, T. Erhardt³, H. Fischer³, A. Landais⁴, J. B. Pedro^{1,5,6}, G.
Vettoretti¹, A. Grinsted¹, V. Gkinis¹, B. Vaughn⁶, A. Svensson¹, B. M. Vinther¹, J. W. C. White⁶

¹ Centre for Ice and Climate, Physics of Ice, Climate and Earth, Niels Bohr Institute, University of
Copenhagen, Tagensvej 16, 2200 Copenhagen, Denmark;

² Université Grenoble Alpes, CNRS, IRD, IGE, 38000 Grenoble, France

³ Climate and Environmental Physics, Physics Institute & Oeschger Center for Climate Change Research,
University of Bern, Sidlerstrasse 5, 3012 Bern, Switzerland;

⁴ Laboratoire des Sciences du Climat et de l'Environnement, LSCE/IPSL, CEA-CNRS-UVSQ, Université
Paris-Saclay, Gif-sur-Yvette, France;

⁵ Australian Antarctic Division, Channel Highway, Kingston, Tasmania;

⁶ Australian Antarctic Program Partnership, Institute for Marine and Antarctic Studies, University of
Tasmania, Hobart, Tasmania;

⁷ Institute of Arctic and Alpine Research, University of Colorado, Boulder, Colorado 80309-0450, USA.

*#Corresponding author: E. Capron, email: emilie.capron@univ-grenoble-alpes.fr*

**SUPPLEMENTARY ONLINE MATERIAL**

**1. Greenland NGRIP and NEEM ice core measurements**

The NEEM water isotope sections are part of the continuous high-resolution water isotope record
covering 8-130 ky b2k¹. Analyses have been performed at the Niels Bohr Institute at the University of
Copenhagen using IR Cavity Ring Down Spectrometry on discrete samples with a resolution of 5 cm. The
combined uncertainty of the measurements (1σ) is 0.05‰ and 0.4 ‰ for $\delta^{18}\text{O}$ and δD , respectively. In the
present study we use and show 600 yr-long sections covering 25 abrupt transitions (Figure S1). In
addition to using the existing NGRIP high-resolution $\delta^{18}\text{O}$ profile², we also present new d-excess data
from the NGRIP ice core for 300-500-yr-long time windows centred on 12 abrupt transitions that were

measured at 5 cm resolution at the Institute of Arctic and Alpine Research (INSTAAR) Stable Isotope Lab
(SIL) (University of Colorado). A first set of measurements were performed in 2006-2007 across the
onsets of GI-3, GI-4, GI-5.2, GI-8c, GI-10, GI-11, GI-12c, GI-18, GI-19.1, GI-19.2 GI-20c and GI-25 using
an automated uranium reduction system coupled to a VG SIRA II dual-inlet mass spectrometer³. A
second set of measurements was performed in 2016-2017 using a Picarro CRDS analyser⁴ in order to fill
some data gaps remaining from the 2006-2007 dataset. In addition, the full section covering GI-18 as well
as 20 depth levels for each other section that were already measured back in 2006 were re-measured
with the Picarro instrument in order to quantify possible offsets between the old and the newer datasets.
Accuracy for new NGRIP $\delta^{18}\text{O}$ and δD measurements using the mass spectrometry-based method is
0.07‰ and 0.5‰ for $\delta^{18}\text{O}$ and δD , respectively, and is 0.1‰ and 1‰ for $\delta^{18}\text{O}$ and δD respectively using
the laser spectroscopy-based method. The water isotope records ($\delta^{18}\text{O}$, d-excess) have a temporal
resolution of better than 1 yr at 10 ka b2k, ~3 yr at 45 ka b2k, ~4 yr at 80 ka b2k and ~5 yr at 105 ka b2k
for NGRIP and of ~1 yr at 10 ka b2k, ~4 yr at 45 ka b2k, ~7 yr at 80 ka b2k and ~18 yr at 105 ka b2k for
NEEM.

The NGRIP and NEEM high-resolution $[\text{Ca}^{2+}]$ and $[\text{Na}^+]$ records over the past 60 ka are published in ref.
5, and in our study, we present the records extended back to ~108 ka b2k. They were measured on both
ice cores using the Continuous Flow Analysis (CFA) system of the University of Bern allowing for an
annual-to-pluri-annual temporal resolution (methodological details are presented in refs. 5,6). The
effective resolution of the CFA records is between 1 and 2 cm and the relative concentration uncertainty
is typically 10%. Here, we use the $[\text{Ca}^{2+}]$ and $[\text{Na}^+]$ records averaged to annual resolution. For the depth
range corresponding to each year in the GICC05 time scale, average impurity concentrations are
calculated directly for NGRIP. For NEEM, the corresponding NEEM depth range is first calculated by
interpolation between the time scale transfer match points of ref. 7, and then the annual average impurity
concentration for that depth range is calculated. The interpolation scheme thus assumes similar
accumulation variability patterns between NGRIP and NEEM.

2. Statistical ramp-fitting analyses of the abrupt transitions

The assumption that the transitions are adequately described by a linear change from one stable state to
another is not trivial and has been challenged previously, but neither our observations nor the current
understanding of the nature of the transitions justify employing a model with more degrees of freedom.
We fit a ramp function (f) to the data in a window around each transition event (\mathbf{Y}), the middle of which is
taken from ref. 8 (Table S1). The ramp function is parametrized by the temporal midpoint t_{mid} , the
transition duration Δt , and the data levels before and after the transition, y_0 and $y_0 + \Delta y$. This formulation
ensures that the ramp parameters are close to mutually independent, which is generally a desirable
property for efficient probabilistic modelling. In addition to the ramp model's four parameters ($t_{\text{mid}}, \Delta t, y_0,$
Δy), we model the residuals as autocorrelated noise. This introduces two additional unknown parameters
for the residual variance (σ^2) and the autocorrelation length (τ). These six unknown model parameters are
arranged into a model vector (\mathbf{m}). The likelihood (L) of a proposed parameter set \mathbf{m} is calculated as the
probability density that the residuals ($\mathbf{Y}-f(\mathbf{m})$) are a realisation of red noise with the proposed noise
characteristics (see ref. 5). Credible parameter ranges are calculated from the posterior probability density
obtained via Bayes' theorem:

$$16 \quad P_{\text{post}}(\mathbf{m}|\mathbf{Y}) \propto L(\mathbf{Y}|\mathbf{m}) \cdot P_{\text{prior}}(\mathbf{m})$$

[revised manuscript text omitted]
[2211-00103.00101](https://doi.org/2210.1175/JCLI-D-2211-00103.00101) (2012).

21 Shields, C. A. *et al.* The low-resolution CCSM4. *Journal of Climate*, [https://doi.org/10.1175/JCLI-](https://doi.org/10.1175/JCLI-D-1111-00260.00261)
[D-1111-00260.00261](https://doi.org/10.1175/JCLI-D-1111-00260.00261) (2012).

22 Argus, D. F., Peltier, W. R., Drummond, R. & Moore, S. The Antarctic component of glacial isostatic
adjustment model ICE-6G_C (VM5a) based upon GPS measurements of vertical motion of the
crust, exposure age dating of ice thickness variations and relative sea level histories. . *Geophys.*
*J. Int.* **198**, 537–563, <https://doi.org/510.1093/gji/ggu1140> (2014).

Peltier, W. R., Argus, D. F. & Drummond, R. Space geodesy constrains ice age terminal
deglaciation: The global ICE-6G_C (VM5a) model. *J. Geophys. Res. Solid Earth* **120**, 450–487,
<https://doi.org/410.1002/2014JB011176>. (2015).

- Vettoretti, G. & Peltier, W. R. Last Glacial Maximum ice sheet impacts on North Atlantic climate
variability: The importance of the sea ice lid. *Geophys. Res. Lett.* **40**, 6378–6383,
<https://doi.org/6310.1002/2013GL058486> (2013).
- Peltier, W. R. & Vettoretti, G. Dansgaard-oeschger oscillations predicted in a comprehensive model
of glacial climate: a "kicked" salt oscillator in the atlantic. *Geophys. Res. Lett.* **41**, 7306-7313.
<https://doi.org/7310.1002/2014GL061413> (2014).
- Vettoretti, G. & Peltier, W. R. Fast Physics and Slow Physics in the Nonlinear Dansgaard–Oeschger
Relaxation Oscillation. *Journal of Climate* <https://doi.org/10.1175/JCLI-D-17-0559.1> (2018).
- Dansgaard, W. Stable isotopes in precipitation. *Tellus* **16**, 436-468 (1964).
- Svensson, A. *et al.* A 60 000 year Greenland stratigraphic ice core chronology. *Climate of the Past*
**4**, 47-57 (2008).
- Li, C., Battisti, D. S., Schrag, D. P. & Tziperman, E. Abrupt climate shifts in Greenland due to
displacements of the sea ice edge. *Geophysical Research Letters* **32**, L19702, doi:
19710.11029/12005GL023492 (2005).
- Sime, L., Hopcroft, P. O. & Rhodes, R. Impact of abrupt sea ice loss on Greenland water isotopes
during the last glacial period. *Proceedings of the National Academy of Sciences* **116**, 4099–4104
(2019).
- Dansgaard, W., White, J. W. C. & Johnsen, S. J. The abrupt termination of the Younger Dryas
climate event. *Nature* **339 (6225)**, 532-534 (1989).
- Masson-Delmotte, V. *et al.* Deuterium excess reveals millennial and orbital scale fluctuations of
Greenland moisture origin. *Science* **309**, 118-121 (2005).
- Vettoretti, G. & Peltier, W. R. Thermohaline instability and the formation of glacial North Atlantic
super polynyas at the onset of Dansgaard-Oeschger warming events. *Geophys. Res. Letters*,
10.1002/2016GL068891 (2016).
- Sadatzki, H., , *et al.* Sea ice variability in the southern Norwegian Sea during glacial Dansgaard-
Oeschger climate cycles. *Science Advances* **5 (3)**, eaau6174 . doi: 6110.1126/sciadv.aau6174
(2019).
- Hurrell, J. W. Decadal Trends in the North Atlantic Oscillation: Regional Temperatures and
Precipitation. *Science* **269**, 676-679 (1995).
- Hurrell, J. W. & Deser, C. North Atlantic climate variability: The role of the North Atlantic Oscillation.
*J. Mar. Syst.* **78**, 28-41 (2009).
- Hurrell, J. W., Kushnir, Y., Ottersen, G. & Visbeck, M. in *The North Atlantic Oscillation: Climatic*
*Significance and Environmental Impact* 1-35 (2003).
- Danabasoglu, G., Landrum, L., Yeager, S. G. & Gent, P. R. Robust and Nonrobust Aspects of
Atlantic Meridional Overturning Circulation Variability and Mechanisms in the Community Earth
System Model. *Journal of Climate*, <https://doi.org/10.1175/JCLI-D-1119-0026.1171> (2019).

Svensson, A. *et al.* A 60 000 year Greenland stratigraphic ice core chronology. *Clim. Past* **4**, 47-
57, doi:10.5194/cp-4-47-2008 (2008).

Laskar, J. *et al.* A long-term numerical solution for the insolation quantities of the Earth. *A&A* **428**,
261–285 (2004).

Bereiter, B. *et al.* Revision of the EPICA Dome C CO₂ record from 800 to 600 kyr before present.
*Geophysical Research Letters* **42**, 542-549, doi:10.1002/2014GL061957 (2015).

Lisiecki, L. E. & Raymo, M. E. Plio-Pleistocene Stack of 57 Globally Distributed Benthic d¹⁸O
Records. *Paleoceanography* **20**, doi:10.1029/2004PA001071 (2005).

Grant, K. M. *et al.* Sea-level variability over five glacial cycles. *Nature Communications* **5**, 5076,
doi:10.1038/ncomms6076 (2014).

NorthGRIP community members. High-resolution record of Northern Hemisphere climate
extending into the last interglacial period. *Nature* **431**, 147-151 (2004).

- 1 **Table S1.** Search intervals used for the ramp-fitting analysis on each abrupt transition covered by ice
- 2 core data and shown in this study.

Transition toward:	Search intervals for ramp-fitting analysis (yr b2k)	
Holocene	11453	11953
GI-1e	14442	14942
GI-2.2	23240	23590
GI-3	27630	28030
GI-4	28650	29150
GI-5.2	32250	32750
GI-7c	35230	35730
GI-8c	37970	38470
GI-10	41210	41710
GI-11	43090	43590
GI-12c	46610	47110
GI-14e	54070	54470
GI-15.1	54920	55320
GI-15.2	55550	56050
GI-16.2	58190	58480
GI-17.1c	58880	59280
GI-17.2	59340	59640
GI-18	63900	64350
GI-19.1	69470	69870
GI-19.2	72090	72590
GI-20c	76190	76690
GI-21.1e	84560	84960
GI-22g	89840	90240
GI-23.1	103840	104240
GI-25a	110700	111190

3

**Table S2.** Ages (yr b2k) of the onset, t2, and end, t1, and equivalent depths (m), d2 and d1 respectively,
 of the studied transitions together with their durations and associated uncertainty intervals (**marginal**
 **posterior** 5-95% credible intervals) found by the ramp-fitting analysis on NGRIP and NEEM ice-core
 tracers. See excel file attached.

**Table S3.** Results of the test regarding the significance of the correlation between NGRIP and NEEM
 transition durations considering seven different groups of data: $\delta^{18}\text{O}$ transitions only, d-excess only,
 $[\text{Ca}^{2+}]$ transitions only, $[\text{Na}^+]$ transitions only, both $\delta^{18}\text{O}$ and d-excess transitions (referred to as Water
 isotope transitions), both $[\text{Ca}^{2+}]$ and $[\text{Na}^+]$ transitions (referred to as Impurity transitions) and finally
 considering the transitions in all tracers together.

Correlation between	N	Frequency $s >s_0$
$\delta^{18}\text{O}$ transitions only	24	0.0311
d-excess transitions only	13	0.0623
Ca^{2+} transitions only	22	0.178
Na^+ transitions only	18	0.0229
Water isotope transitions only	37	0.00055
Impurity transitions only	40	0.008
All transitions	77	0.00003

10

11 **Table S4.** Six modeled abrupt events from three simulations run under different prescribed atmospheric
 12 CO_2 concentrations with the low-resolution version of CCSM4. Search intervals used for the ramp-fitting
 13 analysis on the time series of simulated climate parameters are also indicated.

Transition toward:	Prescribed CO_2 concentrations	Search intervals for ramp-fitting analysis (yr)	
Modeled event 1	185 ppm	3000	3500
Modeled event 2	185 ppm	4750	5250
Modeled event 3	200 ppm	2750	3250
Modeled event 4	200 ppm	5300	5800
Modeled event 5	210 ppm	3100	3600
Modeled event 6	210 ppm	5350	5850

**Figure S1.** High-resolution NGRIP and NEEM ice core records over 600-yr-long windows covering the
 studied abrupt transitions: $\delta^{18}\text{O}$ from NGRIP (dark blue) and NEEM (light blue) (ref. 2, this study), d-
 excess from NGRIP (red) and NEEM (orange) (this study), annually-resampled $\log(\text{Ca}^{2+})$ from NEEM
 (dark green) and NGRIP (khaki green) (ref. 5; this study) and NGRIP annual layer thickness (grey, ref.
 39). Onset points, end points (symbols) and ramps (oblique lines) together with associated uncertainty
 intervals (horizontal shaded lines) found by the ramp-fitting analyses (this study) are indicated. Vertical
 dashed lines indicate the search interval for the ramp-fitting tool.

1

1
2

**Figure S2.** Test of ramp fitting in the presence of autocorrelated noise. 20 different artificial noisy ramps
were analysed with our ramp fitting method and the resulting transition durations for each iteration (open
blue circle) are displayed together with the marginal posterior 5-95% credible intervals (vertical blue bars).
The true duration of 50 years (red dashed line) is within the 5-95% credible intervals in 19 of the 20
cases.

**Figure S3.** Sensitivity tests on the NGRIP $\delta^{18}\text{O}$ transitions into GI-6 (left panels) and GI-12c (right
 panels). The bold blue ramps are defined by the ramp-fitting model run over three search windows of
 different widths: (a, d) ± 100 yr, (b, e) ± 200 yr and (c, f) ± 250 yr. The durations of the resulting transitions
 are indicated in blue and the used search window is indicated by the horizontal black arrows and the
 white boxes.

**Figure S4.** Scatter plots of the NEEM transition duration vs NGRIP transition duration for (a)
 $\delta^{18}\text{O}$ (blue squares), (b) d-excess (red triangles), (c) $[\text{Ca}^{2+}]$ (green dots), (d) $[\text{Na}^+]$ (purple
 diamonds). In e) all tracers ($\delta^{18}\text{O}$ in plain blue triangle, d-excess in opened blue triangle, $[\text{Ca}^{2+}]$
 in plain pink circle and $[\text{Na}^+]$ in open pink circles). Marginal posterior 5-95% credible intervals
 are also indicated (light grey). A 1:1 line is added in dark grey in each graph.

1 **Figure S5.** Duration estimates of the transitions in NEEM $\delta^{18}\text{O}$ and δD into each studied GI.

2

3

4

**Figure S6.** Comparison of the orbital-scale climatic background and the transition durations in NGRIP
and NEEM ice-core tracers. Marine Isotope Stages (MIS) are indicated by the grey bars; MIS boundaries
are defined following Lisiecki and Raymo (2005). (a) 65°N summer insolation (grey, ref. 40) and
atmospheric CO₂ concentration composite record from Antarctic ice cores (green, ref. 41); (b) Benthic
foraminifera δ¹⁸O composite (light blue, ref. 42) and Red Sea Relative Sea Level (RSL) (probability
maximum, dark blue, ref. 43); (c) NGRIP transition durations in δ¹⁸O (blue), d-excess (red), [Ca²⁺] (green),
[Na⁺] (purple) and annual layer thickness (λ, black) superimposed onto the NGRIP δ¹⁸O record (grey, ref.
44). (d) NEEM transition durations in δ¹⁸O (light blue), d-excess (orange), [Ca²⁺] (khaki green) and [Na⁺]
(pink) also superimposed onto the NGRIP δ¹⁸O record (grey).

**Figure S7.** Simulated climatic time series with unforced abrupt oscillations: surface air temperature (blue)
 and precipitation rate (grey) both at the model grid point closest to NGRIP, sea-ice extent in the Irminger
 Seas (brown) and NAO index (pink). Onsets, end points (symbols) and ramps (oblique lines) together with
 associated uncertainty intervals (horizontal shaded lines) found by the ramp-fitting analyses (this study)
 are indicated. Note that the time axis is reversed in order to ease the comparison with ice-core data time
 series.

**Figure S8.** Simulated stadal (a) and interstadial (b) sea-ice concentration under prescribed atmospheric
 CO₂ concentrations of 185 ppm in the low-resolution version of CCSM4.

**Figure S9.** Empirical Orthogonal Function (EOF) analysis on the simulated North Atlantic Oscillation
 (NAO) under prescribed atmospheric CO₂ concentrations of 185 ppm in the low-resolution version of
 CCSM4, with (a) EOF1, (b) EOF2 and (c) EOF3 and the associated explained variance (var). (d-f) Time
 series of the Principal Components 1, 2 and 3 across the unforced abrupt event 1.

REVIEWERS' COMMENTS

Reviewer #1 (Remarks to the Author):

The authors have satisfied my original concerns, and I think that the manuscript is ready for publication with no further edits.

Reviewer #2 (Remarks to the Author):

Review of "The anatomy of past abrupt warmings recorded in Greenland ice" by Emile Capron et al.

I want to thank and commend the authors for their exemplary response to all reviewer comments. Their responses are thorough and insightful, and answer all my comments in a straightforward way. I wish all authors were this responsive to reviewer comments! I fully support publication of this paper.

I have one minor request, which concerns the last sentence of the abstract:

"Our results hint that it may not be possible to infer statistically-robust leads and lags between the different components of the climate system because of their tight coupling".

I think this statement is too general, because they have only shown this for Greenland DO events. For example, I think we can robustly say something about the timing of ice volume and insolation, and other lags/leads in the climate system. Perhaps that is implicit, but I think it would be good to be explicit about it. Also the bipolar phasing of abrupt climate change can be distinguished because it has a much longer timescale (Svensson et al. 2020 doi: 10.5194/cp-16-1565-2020). So perhaps something like:

"Our results hint that it may not be possible to infer statistically-robust leads and lags between the different components of the climate system DURING THE GREENLAND DO CYCLE because of their tight coupling".

Congratulations on a great paper. I look forward to seeing it in print.

Best, Christo Buizert

Below, we provide in blue our answers to the final reviewers' comments
All the best,
Emilie Capron on the behalf of all co-authors.

REVIEWERS' COMMENTS

Reviewer #1 (Remarks to the Author):

The authors have satisfied my original concerns, and I think that the manuscript is ready for publication with no further edits.

> That is great news. Many thanks again for your constructive review.

Reviewer #2 (Remarks to the Author):

Review of "The anatomy of past abrupt warmings recorded in Greenland ice" by Emile Capron et al.
I want to thank and commend the authors for their exemplary response to all reviewer comments. Their responses are thorough and insightful, and answer all my comments in a straightforward way. I wish all authors were this responsive to reviewer comments! I fully support publication of this paper.

I have one minor request, which concerns the last sentence of the abstract:

"Our results hint that it may not be possible to infer statistically-robust leads and lags between the different components of the climate system because of their tight coupling".

I think this statement is too general, because they have only shown this for Greenland DO events. For example, I think we can robustly say something about the timing of ice volume and insolation, and other lags/leads in the climate system. Perhaps that is implicit, but I think it would be good to be explicit about it.

Also the bipolar phasing of abrupt climate change can be distinguished because it has a much longer timescale (Svensson et al. 2020 doi: 10.5194/cp-16-1565-2020). So perhaps something like:

"Our results hint that it may not be possible to infer statistically-robust leads and lags between the different components of the climate system DURING THE GREENLAND DO CYCLE because of their tight coupling".

Congratulations on a great paper. I look forward to seeing it in print.

Best, Christo Buizert

> We are pleased that Christo Buizert is satisfied with our answers to his comments and the accompanying changes we made in the manuscript. Following his last comments, we have rephrased the last sentence of the abstract as such:

*Our results hint that **during these abrupt events**, it may not be possible to infer statistically-robust leads and lags between the different components of the climate system because of their tight coupling.*

Many thanks again for his constructive comments that greatly helped improving our study.